# Ultrahigh energy-dissipation elastomers by precisely tailoring the relaxation of confined polymer fluids

Jin Huang[1], Yichao Xu[1,2], Shuanhu Qi[1,3], Jiajia Zhou [1,3✉], Wei Shi[1], Tianyi Zhao[1] & Mingjie Liu [1,2,3,4✉]

Energy-dissipation elastomers relying on their viscoelastic behavior of chain segments in the glass transition region can effectively suppress vibrations and noises in various fields, yet the operating frequency of those elastomers is difficult to control precisely and its range is narrow. Here, we report a synergistic strategy for constructing polymer-fluid-gels that provide controllable ultrahigh energy dissipation over a broad frequency range, which is difficult by traditional means. This is realized by precisely tailoring the relaxation of confined polymer fluids in the elastic networks. The symbiosis of this combination involves: elastic networks forming an elastic matrix that displays reversible deformation and polymer fluids reptating back and forth to dissipate mechanical energy. Using prototypical poly (n-butyl acrylate) elastomers, we demonstrate that the polymer-fluid-gels exhibit a controllable ultrahigh energy-dissipation property (loss factor larger than 0.5) with a broad frequency range ($10^{-2}$ ~ $10^{8}$ Hz). Energy absorption of the polymer-fluid-gels is over 200 times higher than that of commercial damping materials under the same dynamic stress. Moreover, their modulus is quasi-stable in the operating frequency range.

[1] Key Laboratory of Bioinspired Smart Interfacial Science and Technology of Ministry of Education, School of Chemistry, Beihang University, Beijing, P. R. China. [2] Beijing Advanced Innovation Center for Biomedical Engineering, Beihang University, Beijing, P. R. China. [3] International Research Institute for Multidisciplinary Science, Beihang University, Beijing, P. R. China. [4] Research Institute of Frontier Science, Beihang University, Beijing, P. R. China. ✉email: jjzhou@buaa.edu.cn; liumj@buaa.edu.cn

Vibrations and noises of various frequency bands are ubiquitous in various engineering fields. For instance, vehicles[1], aircrafts[2], and noises[3] are the common vibration sources with diverse associated frequency ranges of $10^0$–$10^4$, $10$–$10^2$, and $10^{-1}$–$10^6$ Hz, respectively. These vibrations can cause malfunctioning, resonance, or fatigue failure of critical structures and human injury[4,5]. To effectively eliminate complex vibration interference, there is an urgent need to develop high-performance damping materials over a broad frequency range. Amorphous polymers with the viscoelastic property are often utilized as conventional damping materials because of the high energy dissipation induced by strong internal friction of chain relaxation in the glass transition region[6,7]. The width of the effective damping region coincides with the glass transition region, which is dominated by the dynamic heterogeneity of chain relaxation[8,9]. Based on this mechanism, strategies of tuning dynamic heterogeneity to broaden the frequency range have been developed, such as adding nanofillers[10–12], blending polymers[13–15], polymer/organic molecules hybrids[16,17], copolymerization[8,18], gradient polymers[19,20], interpenetrating polymer networks[21,22], and polymers with dangling chains[23–25]. However, all these strategies cannot significantly broaden the effective damping region due to the limitation of inherently narrow glass transition regions for general polymer materials, with a frequency range normally spanning $10^2$ Hz[26]. In addition, a large drop in modulus about three or four orders of magnitude is unavoidable in a small temperature variation, inhibiting the practical application[27]. Therefore, it is still challenging to develop elastomers that simultaneously exhibit high damping factor and quasi-stable modulus over a broad frequency range.

In nature, biological damping tissues, like dolphin skins, utilize a synergistic network design by introducing viscous fluids into the dense elastic fiber network to effectively dissipate energy[28]. When the skin vibrates under the impact of water pressure, the viscous fluids flow in and out of the elastic fiber network to effectively eliminate the vibration caused by the water impact. This inspired us to find an alternative approach to modulate the energy dissipation of polymer networks.

Herein, we report high energy-dissipation polymer-fluid-gels (PFGs) over a broad frequency range by infusing viscous polymer fluids into the elastic networks, where polymer fluids repeating back and forth under the imposed stress serve as the dissipative medium. The PFGs' energy-dissipation property can be precisely tailored at desired frequencies. The PFGs exhibit high energy-dissipation performance (loss factor larger than 0.5) over a broad frequency range ($10^{-2}$–$10^8$ Hz), which exceeds typical state-of-the-art damping materials. In shock and vibration tests, PFGs can reduce the impact force up to 85% and dissipate vibration strength by 90%. In addition, the energy absorption of PFGs is over 200 times higher than that of commercial damping materials under the same dynamic stress. PFGs also exhibit desirable expandability and robust fatigue resistance. They can exhibit elongations of 5000% or more and still preserve the original compressive strength after 1000 compressive cycles at 60% strain. PFGs are expected to have various applications in actuators, wearable devices, soft electronics, and soft robotics.

## Results

### The design rationale of high energy-dissipation PFGs.
Traditionally, polymer networks are infused by small molecules to form liquid gels, such as hydrogels or organogels. The infusion of a fluid phase into the polymer networks will significantly enhance the chain relaxation and reduce the internal frictions, leading to inferior damping property[29–31]. Even although these gels can improve the damping performance by using phase transition behavior, this behavior only occurs at certain temperature conditions and makes them mechanically unstable[25]. Yet, this unfavorable mechanism does not mean rejecting the idea of infusing fluid phases. The solution is to infuse polymer fluids (Fig. 1a(iii)) with a high energy dissipation into the elastic networks (Fig. 1a (i)) to form PFGs (Fig. 1a(ii)). The high dissipation of pure polymer fluids is modified and shifted by the polymer network, allowing much more flexible control over the dissipation properties. We explore the symbiosis of this combination with distinct viscoelastic behaviors in the PFGs: under dynamic mechanical stimulation, the cross-linked networks in the high-elastic state offer reversible deformation by the free segment oscillation (Fig. 1b(i)), while the polymer fluids in the viscous state offer high internal friction by the chain reptation (Fig. 1b(iii)). This synergism of elastic-while-viscous attributes represents the distinctive potential of architectural control over damping and mechanical properties of materials (Fig. 1b(ii)). Specifically, owing to the existence of polymer fluids, the materials exhibit a corresponding whole chain reptation behavior in the high-elastic region of polymer networks, leading to a dramatic increase of energy dissipation. By adjusting the relaxation time of polymer fluids by varying their chain length, the energy dissipation can be qualitatively tailored at desired frequencies. Notably, to broaden the high energy-dissipation range, we precisely infuse several polymer fluids with significantly different chain lengths into the polymer network, where polymer fluids with different chain lengths develop hierarchical relaxations.

### Materials synthesis.
To validate this concept, we chose flexible poly (n-butyl acrylate) (PBA) with a low glass transition temperature ($T_g = -56\,°C$) for our design, where at the room temperature the linear and cross-linked PBA is in the flow state and the high-elastic state, respectively. Initially, we synthesized a series of PBA fluids with systematically varied molecular weights by means of atom transfer radical polymerization (ATRP) (see "Methods")[32]. The molecule structures of PBA fluids were characterized by gel permeation chromatography (GPC) and $^1H$ nuclear magnetic resonance (NMR) spectroscopy (Supplementary Fig. 1 and Supplementary Table 1). PFGs were obtained by photo-initiated free-radical polymerization in the prepolymer solution containing n-butyl acrylate monomer, PBA fluid, chemical cross-linker ethylene glycol dimethacrylate, and photo-initiator (2,2-Diethoxyacetophenone) (Supplementary Fig. 2). Mechanical and energy-dissipation properties of PFGs are controlled by three structural parameters: weight fraction of PBA fluid ($\Phi_{PBA\ fluid}$), the weight fraction of cross-linker ($\Phi_c$), and molecular weights of PBA fluids ($M_n$). The stoichiometry of the starting materials is listed in Table 1 and Supplementary Tables 2 and 3.

### High stretchability and fatigue resistance of PFGs.
We first investigated the general mechanical properties of the polymer matrices with and without polymer fluids by the uniaxial tensile and compress tests. The PFG$_{(0.1\%,\ 40\%-5k)}$ can sustain elongations about 5000% or larger (Fig. 2a, b and Supplementary Movie 1), which is about four times as large as that of the PFG$_{(0.1\%)}$ with the same cross-linking density ($\Phi_c = 0.1\%$) (Table 1). Moreover, for the cyclic tensile stress–strain curve with a maximum applied strain of 3000% (Fig. 2c), the PFG$_{(0.1\%,\ 40\%-5k)}$ shows almost full recovery. This ultrahigh stretchability of PFGs can be explained by the reduction of chain entanglements. For conventional linearchain polymer networks, the maximum elongation ($\lambda_{max}$) is scaled as $\lambda_{max} \propto N_x^{1/2}$, where $N_x$ is the degree of polymerization of segments between the chemical cross-linked points ($N_c$) or entanglement points ($N_e$) depending on their

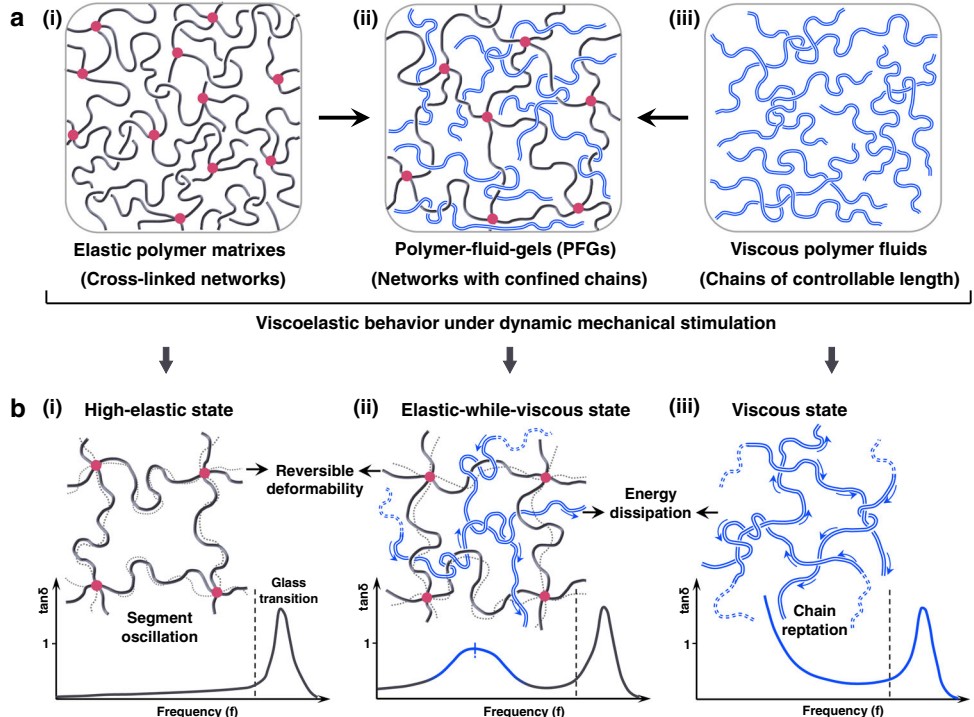

**Fig. 1 Schematic structure of high energy-dissipation polymer-fluid-gels (PFGs) design. a** High energy-dissipation PFGs fabricated by introducing viscous polymer fluids with controlled chain length into the elastic polymer networks. **b** Viscoelastic behavior of polymer networks, PFGs, and viscous polymer fluids under dynamic mechanical stimulation. In the intermediate/low frequencies, polymer networks are in the high-elastic state where their chain segments are free to oscillate. They display considerably low energy dissipation (low tanδ) and highly reversible deformability. In this case, polymer fluids' energy dissipation (tanδ) dramatically increases because of the whole chain reptation. The situation is quite different for PFGs. PFGs are in an elastic state, while viscous, where the whole chain reptation of confined polymer fluids can occur in the high-elastic region of the polymer network.

**Table 1 Molecular parameters and mechanical characteristics of PFGs.**

| Samples | $\Phi_{PBA\ fluid}{}^a$ (%) | $\Phi_c{}^b$ (%) | $M_{n(PBA\ fluid)}$ (kg/mol) | Plateau modulus $G_p{}^c$ (kPa) | Compressive modulus $E^d$ (kPa) |
|---|---|---|---|---|---|
| PFG(3%) | 0 | 3 | / | 132.0 | 453.6 |
| PFG(3%, 20%-35k) | 20 | 3 | 35.1 | 54.9 | 276.2 |
| PFG(3%, 40%-35k) | 40 | 3 | 35.1 | 14.4 | 94.2 |
| PFG(3%, 60%-35k) | 60 | 3 | 35.1 | 4.6 | 18.1 |
| PFG(0.1%) | 0 | 0.1 | 5.2 | / | / |
| PFG(0.1%, 40%-5k) | 40 | 0.1 | 5.2 | / | / |

$^a$Weight fraction of PBA fluids with $M_n = 35$ k.
$^b$Weight fraction of cross-linkers.
$^c$Entanglement plateau modulus characterized by an oscillatory strain rheometry.
$^d$Compressive modulus characterized by stress tests.

magnitude[33]. From the cross-linking density, the molecular weight for a cross-link strand is estimated to be $M_c \cong 10^5$ g mol$^{-1}$, which is larger than the entanglement molecular weight $M_e \cong 10^3$–$10^4$ g mol$^{-1}$ [34]. Thus, the entanglement controls the maximum elongation at break. Since the total number of Kuhn monomers is fixed, the amount of entanglement is inversely proportional to the entanglement length. To increase the elongation at break, we must reduce the total number of entanglements. As a consequence, the PFG(0.1%, 40%-5k) containing abundant unentangled PBA fluids ($M_n = 4$k) exhibits far higher stretchability than the PFG(0.1%). In addition, there are other reasons for the ultrahigh stretchability of the PFG. At the 50-fold extension, some network links break to give this huge extension. This is in line with the 250% residual strain, which points to irreversible chain scission following the 3000% extension (Fig. 2c) of PFG(0.1%, 40%-5k). The other point is that the regular network model is assumed in the theoretical estimation of the maximum

elongation, while the PFG network is irregular. Polymer networks are formed via the free radical polymerization process. Consequently, they possess topological defects, which can improve the tensile rate to some extent.

The inclusion of polymer fluids into a polymer network also leads to other excellent mechanical properties. For example, compared with the liquid gel, the PFG(3%, 60%-35k) displays higher fracture energy, although the mass fraction of polymer networks and cross-linkers of them are the same (Fig. 2d). The PFGs demonstrate high stability (Supplementary Fig. 3) and robust fatigue resistance (Fig. 2e). For instance, the mass of the PFG(3%, 60%-35k) remains almost unchanged at room temperature for 900 h. It still holds original compressive strength after 1000 compressive cycles at 60% strain. These properties should be correlated to the fact that the PFGs are a single-component hybrid with favorable intermolecular van der Waals forces and high compatibility, and the properties of such a hybrid are much

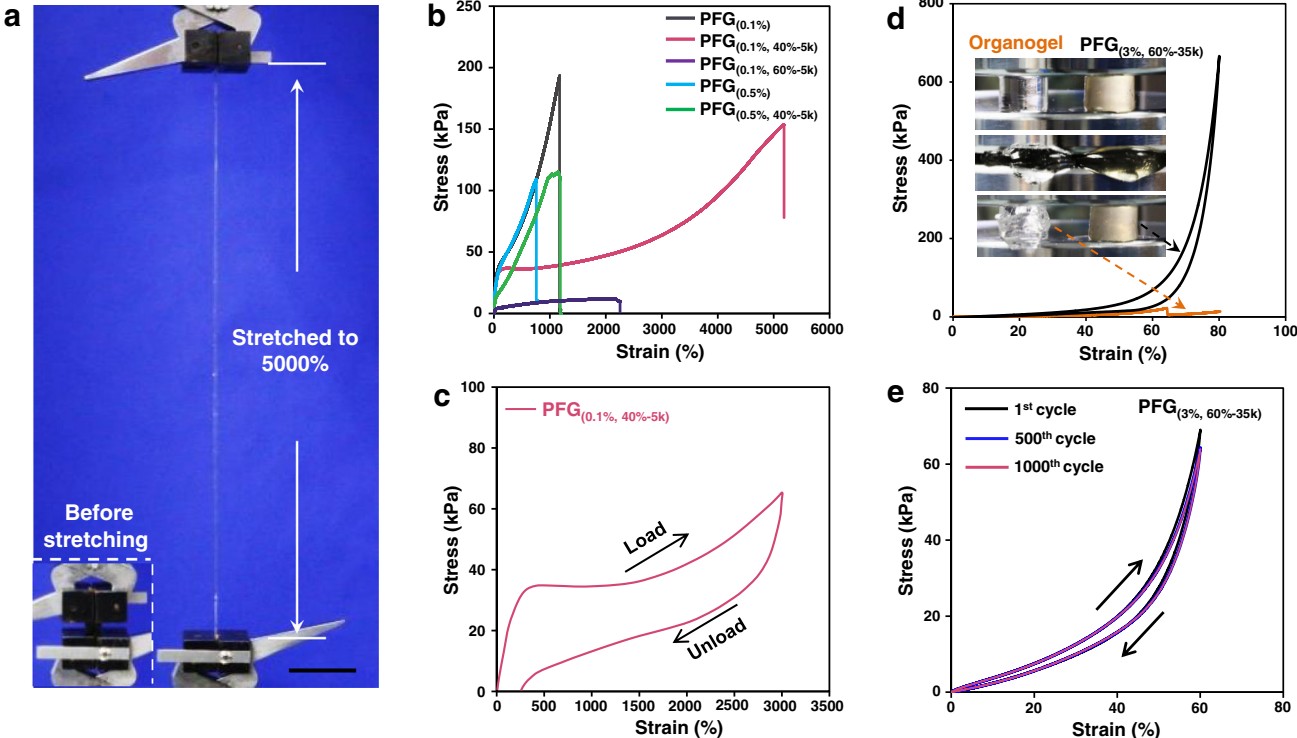

**Fig. 2 Mechanical properties of polymer-fluid-gels (PFGs). a** Photographs of the PFG$_{(0.1\%, 40\%-5k)}$ before and after stretching to 5000%. Scale bars: 5 mm. **b** Tensile stress–strain curves of the PFGs. **c** Cycle tensile of the PFG$_{(0.1\%, 40\%-5k)}$ (up to 3000% strain). **d** Compressive stress–strain curves of the PFG$_{(3\%, 60\%-35k)}$ and the poly (n-butyl acrylate) (PBA) organogel. **e** Cyclic stress–strain curves of the PFG$_{(3\%, 60\%-35k)}$ compressed to 60% strain and released for 1000 cycles.

better than that of its constituents individually (Fig. 2b and Supplementary Fig. 4).

**Dynamic responsive mechanical properties.** In particular, we are interested in the dynamic responsive properties of the PFGs, and we tested such mechanical behavior by using an oscillatory strain rheometer. The PFG is in the linear viscoelastic region within 10% of the strain (Supplementary Fig. 5), which indicates the rheological data measured at 0.5% strain are reversible. Moreover, the dynamic master curves of modulus vs. frequencies for the PFGs were created by time-temperature superposition (TTS), and we utilized Williams–Landel–Ferry (WLF) equation (Supplementary Figs. 6 and 7) and van Gurp–Palmen plots diagram (Supplementary Fig. 8) to verify that PFGs obey TTS.

The PFGs' rheological master curves of frequency ($\omega$) dependence of the storage modulus ($G'$), loss modulus ($G''$), and loss factor (tan$\delta$) are presented in Fig. 3a, b. The tan$\delta$ is a key parameter to characterize the energy-dissipation property of materials[35]. During the measurement, the weight fraction of the cross-linker was kept constant at $\Phi_c = 3\%$, and the content of PBA$_{270}$ fluids ($\Phi_{PBA\ fluid}$) was gradually changed (Table 1). The $G'$ of all the PFGs shows a platform in the low-frequency region ($\omega \rightarrow 0$). The reason is that the $G'$ of the crosslinked network in the PFGs changes slightly, due to the existence of the crosslinks to prevent the network chains slip[27]. As the $\Phi_{PBA\ fluid}$ increases to 60%, the plateau modulus ($G_p$, $\omega \rightarrow 0$) of PFGs decreases from 132 to 4.6 kPa, due to the fact that the existence of polymer fluids immensely reduces the entanglement density of the network matrix (Fig. 3a and Table 1). The compression modulus ($E$) of PFGs shows a similar trend, decreasing from 453.6 to 18.1 kPa (Supplementary Fig. 9 and Table 1). Meanwhile, as $\Phi_{PBA\ fluid}$ increases, the peak of tan$\delta$ for PFGs becomes more evident in the

rubbery region of the network matrix ($\omega$ ~158 rad s$^{-1}$) (Fig. 3b). It arises from the internal friction of the whole chain reptation of polymer fluids when the time scale of this relaxation is comparable to the inverse of the frequency of the applied oscillatory shear. The temperature-dependence rheological curves of PFGs display similar characteristics as well (Supplementary Fig. 10). These results validate that the whole chain reptation of polymer fluids plays a dominant role in the energy dissipation of PFGs.

To analyze the effect of polymer fluid molecular weight ($M_n$) on the tan$\delta$ of PFGs, we synthesized a series of PBA fluids with systematically varied $M_n$. Without the polymer network, Fig. 3c presents the master curves of the PBA fluids with various $M_n$ at 25 °C. There is no entanglement plateau in low $M_n$ PBA fluids ($M_n = 5$, 20, and 35 k), whose curves show viscous flow behavior. In contrast, the higher $M_n$ PBA fluids ($M_n$ larger than or equal to 58 k) exhibit a rubbery plateau region at $G$ ($G_e$) ~1.07 × 10$^5$ Pa, where $G_e$ is the modulus due to entanglements. The modulus is calculated by the van Gurp Palmen method and the minimum tan$\delta$ method[34] (Supplementary Fig. 11). With the increase of $M_n$ (PBA fluid), the breadth of the entanglement plateau increases and the flow transition shifts to a lower frequency region, indicating that the whole chain relaxation time ($\tau$) of PBA fluids becomes longer. With the polymer network, in the PFGs, the relaxation behavior of polymer fluids is characterized by the shift of the peak position of tan$\delta$ (Fig. 3d). With increasing $M_n$ (PBA fluid), the peak value of tan$\delta$ decreases. It is due to the fact that the reptation time of polymer fluids becomes longer, indicating that their reptation occurs at lower frequencies or higher temperatures. In this case, local chain interactions become lower, leading to a decrease in the frictional losses of their reptation. As the $M_n$ (PBA fluid) increases from 5 to 195 k, the frequency corresponding to the peak of tan$\delta$ shifts from ~10$^4$ to ~10$^{-2}$ rad s$^{-1}$, which spans six orders of

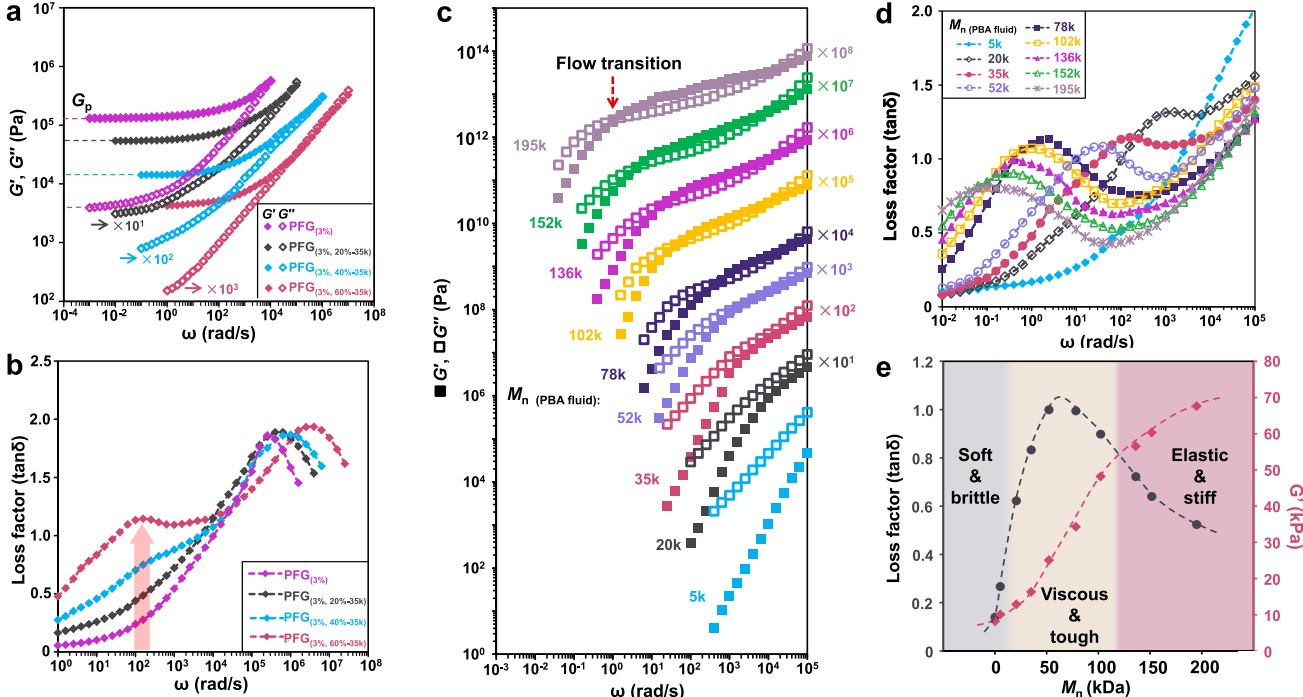

**Fig. 3 Dynamic mechanical master curves of polymer-fluid-gels (PFGs). a, b** Frequency dependence of storage modulus ($G'$), loss modulus ($G''$), and loss factor (tan$\delta$) for PFGs. The master curves were obtained by time-temperature superposition (TTS) and shifted horizontally by the indicating factors for clarity. Plateau modulus ($G_p$) of the PFGs reduces with increasing poly (n-butyl acrylate) (PBA) fluid content. In the high-elastic region of the bulk network, a definite peak of tan$\delta$ becomes more obvious as $\Phi_{PBA\ fluid}$ increases, indicating that the peak originates from the whole chain reptation of PBA fluid. **c** Vertically shifted modulus master curves of PBA fluids with varying $M_{n\ (PBA\ fluid)}$. Filled squares: $G'$, open squares: $G''$. **d** Master curves for the tan$\delta$ of PFGs with $\Phi_c = 3\%$, $\Phi_{PBA\ fluid} = 60\%$, and varying $M_{n\ (PBA\ fluid)}$. As $M_n$ of polymer fluids increases, the definite peak of tan$\delta$ shifts to the lower frequency side. **e** Tan$\delta$ and $G'$ of PFGs with $\Phi_c = 3\%$ and $\Phi_{polymer\ fluid} = 60\%$ as a function of $M_{n\ (PBA\ fluid)}$ at a frequency of 10 rad s$^{-1}$ and $T = 25$ °C. $M_{n\ (PBA\ fluid)}$: molecular weight of PBA fluids, $\Phi_c$: weight fraction of cross-linker, $\Phi_{polymer\ fluid}$: weight fraction of PBA fluid.

magnitude. Based on these results, one can design PFGs with maximum energy dissipation at the desired frequency by properly choosing the relaxations of the polymer fluids.

To understand the mechanism of such a design, we identified the mechanical behavior of PFGs with respect to $M_{n\ (PBA\ fluid)}$ as three characteristic zones in the curves of the tan$\delta$ and $G'$ of PFGs ($\Phi_c = 3\%$, $\Phi_{PBA\ fluid} = 60\%$ and $\omega = 10$ rad s$^{-1}$) (Fig. 3e and Supplementary Fig. 12). In Zone 1, the tan$\delta$ and $G'$ of PFGs with low $M_{n\ (PBA\ fluid)}$ (smaller than 9 k) are less than 0.4 and 10 kPa, respectively, since the chains in PBA fluids are too short to be entangled and their whole chain relaxation has already occurred at high frequencies ($\omega$ larger than $10^4$ rad s$^{-1}$) (Fig. 3d). In this case, PFGs are soft and brittle, and their mechanical properties are that of the liquid gels, like fragile jellies. When the $M_{n\ (PBA\ fluid)}$ is given in the range of 16 to 121 k, the whole chain relaxation of the PBA fluids in the PFGs occurs at frequencies of 10 to $10^3$ rad s$^{-1}$ (Fig. 3d, e). In Zone 2, the $G'$ and tan$\delta$ of the PFGs show dramatic increases, and the PFGs display high energy dissipation and toughness. In Zone 3, when $M_{n\ (PBA\ fluid)}$ larger than 121 k, PBA fluids are heavily entangled and diffuse too sluggishly to relax, where most of PBA fluids are in the rubbery state at $\omega = 10$ rad s$^{-1}$. The increase of $G'$ tends to flatten, and the value of $G'$ approaches the entanglement modulus ($G_e \sim 1.07 \times 10^5$ Pa), while tan$\delta$ gradually declines to 0.52. As a consequence, we can adjust the relaxation time of polymer fluids in the polymer network to obtain PFGs with both optimal damping and mechanical properties.

**Theoretical analysis of the relaxation of confined polymer fluids.** The control of dissipation properties of PFGs can be

analyzed theoretically. According to the reptation model, the relaxation time $\tau$ of polymer fluid chains in a fixed network depends on the cubic power of the molecular weight of polymer fluids[36]. Incorporating the correction due to the contour length fluctuation, the relaxation time is given by Eq. (1).

$$\tau = \frac{b^2 \zeta_0 M_n^3}{M_x M_0^2 \pi^2 k_B T} \left[ 1 - 1.3 \left( \frac{M_x}{M_n} \right)^{0.5} \right]^2 \quad (1)$$

$b$ is the Kuhn length, and $\zeta_0$ and $M_0$ are the frictional coefficient and molecular weight of the monomer, respectively. $M_n$ is the molecular weight of the polymer fluids and $M_x$ is the average molecular weight of cross-linking strands in the network. In the PFGs, since the molecular weight of cross-linking strands ($M_x = 4.2$ kDa) is less than the entanglement molecular weight ($M_e \cong 20$ kDa)[34], the mesh size is less than the tube diameter. In this case, the movement of the polymer fluid chains is restricted by a smaller mesh size. The gray line is a fit of Eq. (1) in Fig. 4a. When the molecular weight of the infused polymer is larger than the network size ($M_x = 4.2$ kDa), the fitted curve shows good agreement with the experimental data $\tau_{ex}$ ($\tau_{ex} = 2\pi/\omega$, where $\omega$ is the value of frequency at the peak of tan$\delta$ curves in the Fig. 3d) (Fig. 4a). The appropriate molecular weight of the polymer fluid obtained by reading the molecular weight in the fitted curve can help developing high energy dissipation at specified frequency $\omega$ for the PFGs.

**Design of high damping PFGs over a broad frequency range.** The basic idea of controlling the dissipating behaviour of the PFGs with a single-component polymer fluid at a localized frequency can be extended to the design of high energy-dissipating

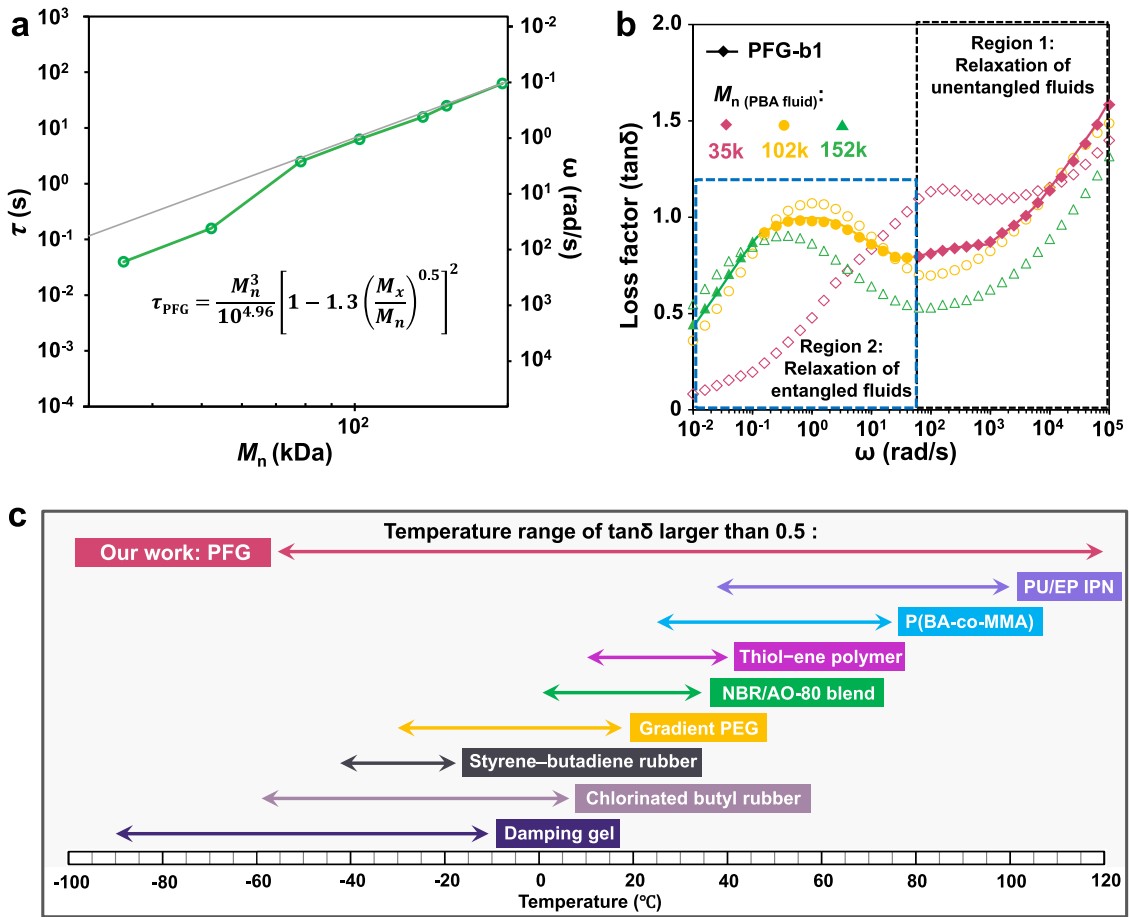

**Fig. 4 Precise macromolecular design of high damping polymer-fluid-gels (PFGs) over a broad frequency range. a** The reptation model predicts the whole chain reptation time ($\tau$) of poly (n-butyl acrylate) (PBA) fluids in a fixed network. **b** Frequency dependence of loss factor (tan$\delta$) at 25 °C for PFG-b1 containing three different molecular weights ($M_{n\,(PBA\,fluid)}$) PBA polymer fluids of a weight fraction of $\Phi_{PBA\,fluid} = 60\%$. In the high-elastic region of the bulk network, multiple distinct peaks of tan$\delta$ can be observed, which demonstrates that the whole chain motion of polymer fluids takes place stepwise. **c** The temperature range of tan$\delta$ larger than 0.5 of the PFGs compared to the reported damping materials (see Supplementary Table 4 for data).

materials with a wide frequency range by infusing several polymer fluids with significantly different chain length into a polymer network. The whole chain relaxation of polymer fluids can take place stepwise, leading to a series of corresponding tan$\delta$ peaks in the rubbery region of the PFGs. The resulting PFGs exhibit high damping performance over a broad frequency range. As a proof-of-principle demonstration, we synthesized a PFG that contains a polymer network with three different PBA fluids (with the $M_n$ of [35 k]/[102 k]/[152 k] in a mass ratio of 2:3:1) (Fig. 4b, Supplementary Figs. 13 and 14 and Supplementary Table 3). Mechanical tests exhibit considerably a high value of tan$\delta$ (larger than 0.5) over a broad frequency range ($10^{-2}$–$10^8$ rad s$^{-1}$) (Supplementary Fig. 13). The tan$\delta$ curve of the PFG-b1 clearly displays the relaxation behaviour of each PBA fluid component, consistent with our design concept (Fig. 4b). Compared to the PFG with a single polymer fluid, the damping performance of PFG-b1 exhibits higher stability and controllability, though the high damping (tan$\delta$ larger than 0.5) frequency range of the PFG with a single polymer fluid (such as $M_n = 102$ k) is close to that of the PFG-b1 under certain circumstances (Fig. 4b). Owning to the gradual relaxation of various PBA fluids in the PFG-b1, the tan$\delta$ curve of PFG-b1 displays a more stable trend rather than a pronounced peak (Fig. 4b), and the PFG-b1 can achieve more possibilities for the damping performance by regulating the PBA fluid ratio. The PFG-b1 also exhibits high tan$\delta$ (larger than 0.5) over a broad temperature range, and the breadth of this region ranges from

−50 °C to more than 120 °C (Supplementary Fig. 15). Furthermore, in the high damping region (tan$\delta$ larger than 0.5), the change of storage modulus ($G'$) of the PFG-b1 is relatively gentle (Supplementary Figs. 13 and 15), which is in contrast to the conventional damping materials based on the glass transition. The viscoelastic properties of PFG-b1 are insensitive to both the temperature and the frequency, thus offer a great opportunity for practical applications. Notably, to our knowledge, the temperature range of tan$\delta$ larger than 0.5 of the PFGs is wider than state-of-the-art damping materials (Fig. 4c and Supplementary Table 4).

**Demonstration of shock absorption and vibration damping.** In order to measure the energy dissipation performance of the PFGs, we systematically tested their vibration and shock absorption properties. We applied a periodic sinusoidal alternating stress (15 kPa) to the PFG$_{(3\%,\,60\%\text{-}35k)}$. The stress–strain curves are elliptic loops defined as hysteresis loops, indicating that the PFG$_{(3\%,\,60\%\text{-}35k)}$ is still in the linear viscoelastic region under 50% strain (Fig. 5a)[37]. Energy dissipation ($\Delta W$) in each cycle can be calculated by the area of the hysteresis loop (Supplementary Fig. 16). As shown in Fig. 5b, the energy dissipation ($\Delta W$) of the PFG$_{(3\%,\,60\%\text{-}35k)}$ is over 200 times higher than that of commercial rubbers under the same dynamic stress (15 kPa). It is worth mentioning that the storage modulus ($G'$) reduces by around 20% after 50 k cycles under 50%

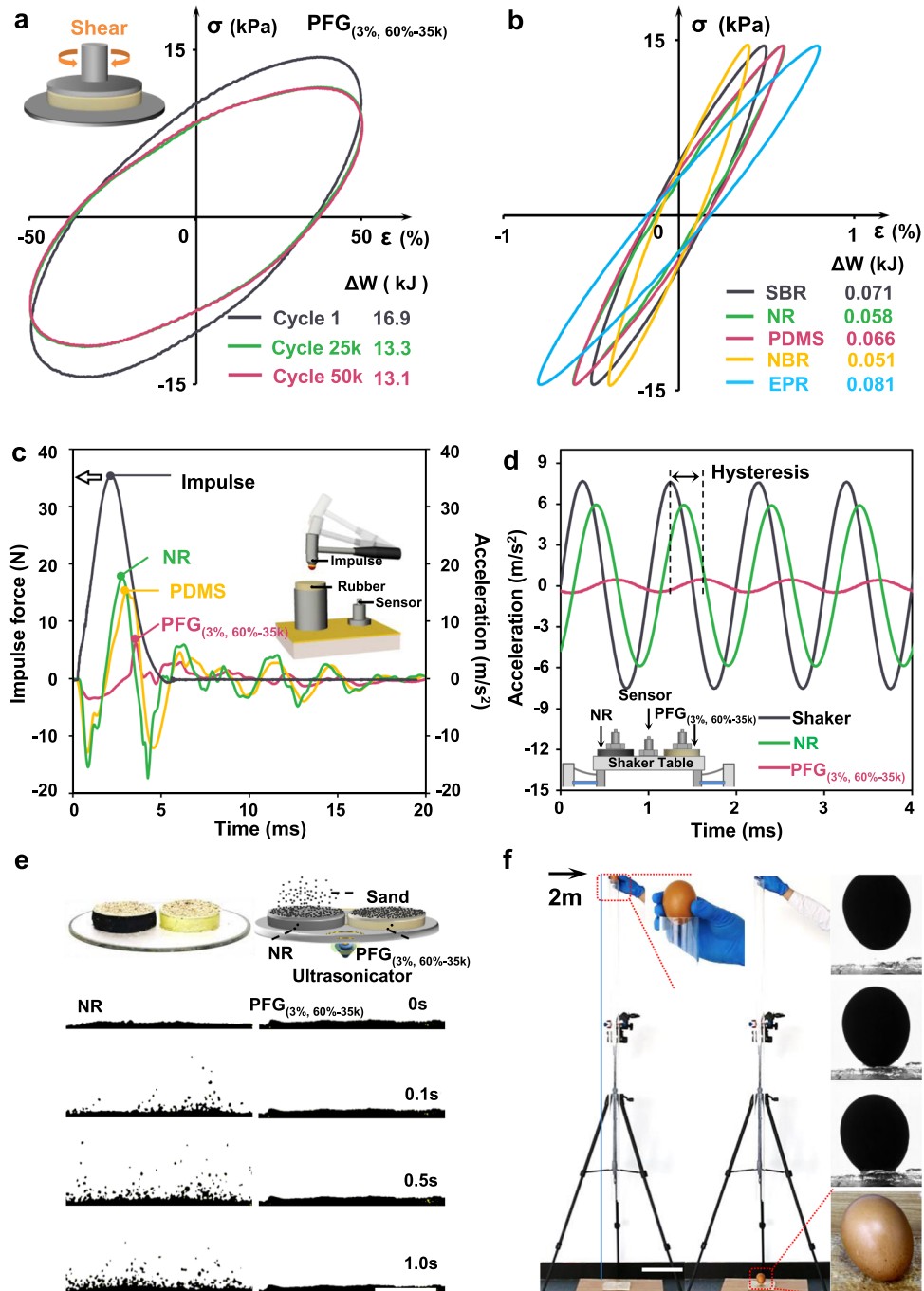

**Fig. 5 Shock absorption and vibration damping of polymer-fluid-gels (PFGs). a, b** Large amplitude oscillatory shear (LAOS) tests of damping materials. The hysteresis loops are elliptical, indicating that PFG(3%, 60%-35k) is located in the linear viscoelastic region with 50% strain. The hysteresis loop area represents mechanical energy converted to heat during each cycle and the area is reduced by around 20% after 50 k cycles. The energy dissipation ($\Delta W$) of the PFG(3%, 60%-35k) is over 200 times higher than that of commercial damping materials under the same dynamic stress. **c** Drop weight impact tests on selected damping materials. The PFG(3%, 60%-35k) can reduce the impact force up to 85%. **d** Shaking table demonstrative experiments of selected damping materials. The amplitude of primitive vibration signal on the PFG(3%, 60%-35k) was dissipated by 10 times; meanwhile, the phase of signal lagged significantly. **e** Acoustic absorption experiments of damping materials. The PFG(3%, 60%-35k) can effectively attenuate sound waves. **f** The egg was dropped from 2 m height onto a 5 mm-thick PFG(1%, 60%-35k) pad and remained unbroken. SBR styrene-butadiene rubber, NR natural rubber, PDMS polydimethylsiloxane rubber, NBR nitrile butadiene rubber, EPR ethylene propylene rubber. Scale bars: 5 mm (**e**), 20 cm (**f**).

strain, further confirming the excellent fatigue resistance of the PFG(3%, 60%-35k) (Supplementary Fig. 17). To verify the energy absorption ability of the PFG, a drop-weight impact test was conducted (Fig. 5c and Supplementary Fig. 18). An impulse force of 35 N was applied to the PFG(3%, 60%-35k) by a drop weight. The results indicate that the PFG(3%, 60%-35k) can reduce the impact force

up to 85% in 0.02 s. Compared to other commercial damping materials like silicone rubber and damping gel, the PFG(3%, 60%-35k) also exhibits a lower rebound. To visualize the impact absorption ability of the PFGs, a raw egg was dropped from 2 m height onto a 5 mm-thick PFG(1%, 60%-35k) pad without any cracks (Fig. 5f, Supplementary Fig. 19, Supplementary Movie 2 and Movie 3), while

dropped onto the natural rubber pad and damping gel with serious rupture (Supplementary Movie 4 and Movie 5). Moreover, the vibration absorption capacity of the PFG$_{(3\%, 60\%-35k)}$ was measured by shaking table demonstrative experiments (Fig. 5d and Supplementary Fig. 20). The amplitude of the original vibration signal with the frequency of 1000 Hz applied to the PFG was dissipated by 90%; the phase lag of the vibration signal was significant. Meanwhile, we tested the acoustic sound absorption of the PFG$_{(3\%, 60\%-35k)}$ using an ultrasonic generator (Fig. 5e, Supplementary Movie 6 and Movie 7), since the acoustic absorption capacity is also an important parameter quantifying the performance of damping materials. When the ultrasound wave with a frequency of 1 MHz is applied, the sands atop the natural rubber (NR) and damping gel spacer beat violently. In contrast, the sands atop the PFG$_{(3\%, 60\%-35k)}$ spacer stay still, suggesting that the acoustic energy is completely dissipated and converted to internal energy.

## Discussion

We have developed materials that provide ultrahigh-energy dissipation over a broad frequency range by infusing polymer fluids into the polymer network. The energy dissipating mechanism is due to the internal friction arising from the whole chain motion of polymer fluid chains in the network matrix. By precisely regulating the relaxation time of polymer fluids, the material can present optimal energy dissipating and mechanical properties at the desired frequency. Furthermore, the general idea is exploited for the design of a high energy-dissipation property over a wide frequency range through infusing several polymer fluids with significantly different chain lengths into the matrix. Notably, the modulus of this material is quasi-stable in the corresponding frequency range, unlike conventional damping materials. In addition, the material also exhibits superior stretchability and fatigue resistance. Therefore, we anticipate that this design concept will provide a general approach to the development of advanced energy-dissipation materials with broad practical applications.

## Methods

**Materials**. Butyl acrylate (BA, 99%, Aladdin) was purified using a basic alumina column. Copper(I) chloride (CuCl, 99.9%, Alfa) was purified according to the reported method[38]. CuCl (5 g) was stirred in glacial acetic acid (100 mL) overnight. The content was filtered through a Buchner funnel and washed three times with ethanol and diethyl ether, dried in a vacuum overnight. All other reagents (Ethylene glycol dimethacrylate (EGDMA, 99%), 2,2-Diethoxyacetophenone (98%), Ethyl 2-bromoisobutyrate (EIBB, 98%,), tris[2-(dimethylamino)ethyl]amine (Me6TREN, 99%), 2,2-Diethoxyacetophenone (98%), toluene, and tetrahydrofuran) were purchased from Aldrich and used as received.

**Synthesis of PBA fluids**. The PBA fluids were synthesized via ATRP[29]. Toluene (50 ml) was added to a 100 ml flask equipped with a stir bar and bubbled using dry nitrogen for 1 h. EIBB (initiator), Me$_6$TREN (ligand), CuCl (catalyst), and BA (monomer) were quickly added under nitrogen with the molar ratio of EIBB/Me$_6$TREN/CuCl/BA of 1:1:1:$n$. The polymerization reaction was carried out at 80 °C for 12 h. The PBA fluids were purified by being precipitated into cold methanol.

**Synthesis of PFGs**. All PFGs were prepared by photo-initiated radical polymerization (Supplementary Fig. S1). The initial reaction mixtures contained: monomer (butyl acrylate, BA), PBA fluid, cross-linker (ethylene glycol dimethacrylate, EGDMA), and 0.5% photoinitiator (2,2-Diethoxyacetophenone). Then, the mixtures were added to a Teflon mould and polymerized at room temperature for 2 h using an ultraviolet lamp (365 nm). The resulting PFGs were dried for one week in a vacuum of 100 °C to remove unreacted monomers.

**Molecular structural characterization**. $^1$H NMR spectra were recorded at 22 °C on a Bruker Avanced III spectrometer operating at 400 MHz in CDC1$_3$, and the chemical shifts were determined with tetramethylsilane as the internal reference. GPC was conducted using a Waters 2414 equipped with a refractive Index Detector at 35 °C; the eluent was absolute tetrahydrofuran.

**Mechanical tests**. Mechanical tensile-stress experiments were carried out by a SUNS UTM4000 instrument at room temperature (25 °C). For compressive stress–strain tests, the PFGs were cylinders with 12 mm diameter × 10 mm height at a displacement rate of 10 mm/min. For tensile stress–strain tests, the PFGs were cylinders with 6 mm diameter × 30 mm height at a displacement rate of 20 mm/min.

**Rheological tests**. The dynamic viscoelasticity of the PFGs was measured by an Anton Paar model MCR-302 rheometer. The samples were placed under a 15 mm-diameter parallel plate. In the strain sweep tests, the shear strain ($\gamma$) was from 0.01 to 10% at the frequency of 10 rad s$^{-1}$ and temperature of 25 °C. In the frequency sweep tests, the angular frequency ($\omega$) was from 0.1 to 100 rad s$^{-1}$ at specified temperatures with the shear strain ($\gamma$) of 0.5%. The master curves at a reference temperature of 25 °C were scaled by TTS. In the temperature sweep measurements, the temperature ($T$) was varied from −120 to 200 °C (2 °C min$^{-1}$) at a frequency of 10 rad s$^{-1}$ with the shear strain of 0.5%.

**Time–temperature superposition (TTS)**. The WLF equation[39] and van Gurp–Palmen plots diagram[40] were utilized to verify that PFGs obey TTS.

*WLF equation*. $G'$ values were measured over a $10^1$–$10^2$ Hz frequency range at different temperatures (−35 to 85 °C, $\Delta T = 10.0$ °C, where $\Delta T$ describes the temperature interval). Supplementary Fig. 6a shows the behaviour of $G'$ at different temperatures as a function of frequency. The master curve obtained by shifting the data in Supplementary Fig. 6a horizontally along the frequency axis with a reference temperature ($T_{ref}$) of 25 °C (Supplementary Fig. 6b). The temperature-dependent shift factors ($\alpha_t$) were calculated from the data in Supplementary Fig. 6b. The shift factors at 55 and −5 °C was chosen to calculate the empirical fitting constants ($C_1$, $C_2$) of the WLF equation, and the shift factors were fitted to the WLF equation (Supplementary Fig. 7).

*Van Gurp–Palmen plots*. The loss angle $\delta$ vs. the complex modulus ($G^*$) was plotted (Supplementary Fig. 8). This way of plotting eliminates the effect of shifting along the frequency axis and yields temperature-independent curves. When the plots are continuous, it indicates that TTS holds.

**Large amplitude oscillatory shear (LAOS) tests**. The LAOS tests of the materials with 5 mm height were measured by an Anton Paar model MCR-302 rheometer at 25 °C. When sinusoidal alternating stress ($\sigma = \sigma_0 \sin \omega t$) was applied to the PFGs, the corresponding strain ($\varepsilon = \varepsilon_0 \sin(\omega t - \delta)$) of PFGs occurred. The $\delta$ was the lagging angle consistent with the $\delta$ of loss factor (tan$\delta$). The hysteresis loop area represents the mechanical energy that converts to heat each cycle[37]. Energy dissipation ($\Delta W$) is calculated through Supplementary Eqs. (2)–(5).

**Drop hammer impact tests**. The impulse signals (the impulse force) of the materials with 5 mm were produced by an impact hammer (PCB Piezotronics Inc.) and the impulse force was in the range of 30–40 N. The response signals were collected by an acceleration sensor.

**Shaking table demonstrative experiments**. The acceleration sensors were placed on the shaking table and materials, respectively. They were all submitted to a sinusoidal longitudinal vibration of 1 kHz frequency and collected response signals.

**Acoustic absorption experiments**. The damping materials with 5 mm height were placed under the Petri dishes containing sands and put on an ultrasonic generator. The frequency of the ultrasonic generator was 1 MHz.

## Data availability

All data is available in the main text or supplementary materials. The data that support the findings of this study are available from the corresponding authors on reasonable request.

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

## Acknowledgements

This research was supported by the National Key R&D Program of China (2017YFA0207800), the National Natural Science Funds for Distinguished Young Scholar (21725401), the National Natural Science Foundation (21774004), the 111 projects (B14009) and the Fundamental Research Funds for the Central Universities.

## Author contributions

M.L. contributed to the initiating idea and J.H. performed all the experiments. J.Z. and S.Q. contributed to the analysis of mechanical properties and the theoretical analysis of the relaxation of polymer fluid. Y.X., W.S., and T.Z. designed the demonstrative experiments of energy absorption. The paper was written by J.H., J.Z. and M.L., and edited by all the authors. All the authors analyzed the data of experiments.

## Competing interests

The authors declare no competing interests.
