## [Peer Review File · Nature Communications]

Reviewers' Comments:

Reviewer #1:

Remarks to the Author:

Report on ms. NCOMMS-20-30498 by Huang et al.

This is an interesting, comprehensive, and quite clearly-described study where the authors propose a novel concept for vibration-damping materials based on gels infused with free polymers (in their liquid state), where the gel network chains and the free polymer are based on identical monomers (poly (n-butyl acrylate) (PBA) in the present study). I like the conceptual novelty of tailoring the relaxation characteristics of the free chains together with the gel-network characteristics and the use of multiple free-chains of different molecular weights to “fine-tune” the damping properties of the composite gel. The results are impressive. There are a number of issues, largely of clarification, that the authors should address, as detailed below.

1. Polymer-infused gels have been described in the literature, e.g. Mu et al., *Extreme Mechanics Letters* 38 (2020) 100742; Osaheni et al., *Acta Biomater.* 46, (2016) 245–255.; *Clin. Biomech.* 67 (2019) 15–19. These should be related to.
2. Soft gels have been used for vibration damping. In particular, a commercial vibration damping gel (I believe there may be others), <https://taica.co.jp/gel/en/about/>. This appears to be a silicone gel imbued with low M siloxane oligomers. This should be noted, and also see below in connection with comparisons.
3. I found it made hard reading constantly to refer to the Supplementary Material, especially when referring to different samples. Thus (line 106) PFG-6 and PFG5 are compared but not clear what they are without chasing up the supplementary. At least the samples should all be in a single table in the main text.
4. I have difficulty understanding the origins of the 50-fold (5000%) elongation (l. 107). The maximal elongation goes like $N^{1/2}$, where N is the number of Kuhn steps in the chain being stretched; but the total number of monomers N_{monomer} either in an entanglement length (103 - 104 Da) or of a cross-link strand (105Da) yield $N_{\text{monomer}} \approx 800$ at most (given that a monomer molecular weight is ca. 120 Da), and even assuming each monomer was a Kuhn step (which it isn't! the number of Kuhn steps is much smaller), then $N^{1/2} < \text{ca. } 30$; not 50. I wonder if at 50-fold extension some network links break to give this huge extension? This would be in line with the 250% residual strain (which by the way is quite a large residual strain and points to irreversible chain scission) following the 3000% extension (fig. 2c) of PFG-6. The authors should comment on this.
5. I am not sure I understand the point about reducing the number of entanglements (l.118 – 121). Is this not a topological constant in the case of cross-linked networks? This point should be clarified.
6. Likewise, please clarify why “the properties of such a hybrid are much better than the properties of its constituents individually” (l.130)
7. Are the rheological data reversible (in view of the hysteresis in fig. 2c)?
8. Not clear how the PFG can be “soft and brittle” at the same time? (l. 171) Please clarify.
9. Comparison with other vibration damping materials: I am not sure that comparison with bulk rubbers is fully appropriate. I would be happy to see comparison with other gels, especially the existing anti-vibration gel, see comment 2 above.
10. The egg-dropping demonstration is very interesting. But they must specify if the egg was raw (liquid inside the shell) or hard-boiled (solid throughout). If it was liquid then indeed this is a truly remarkable demonstration. If solid, somewhat less so... As above, it would be interesting to compare with what happens to the egg on different damping materials including rubbers and the silicone gel mentioned in 2 above.
11. The comparison of the sand experiments should again be made with a vibration-damping gel if possible rather than with natural rubber (see 2 above).

Overall I feel the idea of regulating the relaxation times of the PFGs by different polymeric fluids and their combinations, and implementing this idea to create vibration-damping materials with remarkable properties, is conceptually-novel, leads to impressive results, and is worthy of publication in *Nature Communications*. I will be prepared to review a revised version where the

above points are addressed.

Minor points:

- a. I think they use the term 'disperse phase' (l. 70 and elsewhere) incorrectly. It generally applies to colloidal dispersions, where the disperse phase is the particles, and the continuous phase the surrounding (generally fluid) medium. Why not simply 'liquid phase' or 'fluid phase'?
- b. There is some error in the x-axis of fig. 2j – the $M_n^{1/2}$ values cannot be 3k etc.
- c. At a number of points wad/s appears instead of presumably rad/s (?)
- d. L. 188 – which grey line?

Reviewer #2:

Remarks to the Author:

This manuscript by Huang et al. presents a methodology to create elastomers with tunable damping characteristics. The approach relies on filling a crosslinked polymer network, that serves as the load bearing network, with a polymer melt that aids in energy dissipation. The resulting "filled" elastomers exhibit excellent stretchability and energy dissipation characteristics. In the design aspect, these materials are not different from interpenetrating network elastomers. The innovation, as the authors portray, comes from employing polymers with three molecular weights in the polymer melt component, such that the energy dissipation can be expanded over an extended frequency range. In terms of damping and shock absorbing performance, the demonstrated materials are excellent, although a comparison of the damping performance with contemporary elastomers would have helped the authors convey this point better. Even with the interesting material performance, the publication of this manuscript in Nature Communications cannot be recommended for the following reasons:

1. The authors employ time-temperature superposition (TTS) to arrive at the frequency response of their materials. These frequency sweeps form the basis of a lot of data presented in figures 2 and 3. However, since the materials are a combination of a network and a polymer melt, the applicability of the TTS approach is questionable. The authors do not report the shift factors for TTS and their behavior for the different materials.
2. The authors also do not discuss the frequency sweeps that they report in figure 2f. Why do G' and G'' plateau at low ω ? What is the dependence of both G' and G'' in the high ω region, and why? Lastly, the frequency sweeps (figure 2f) should be reported for the same frequency range as $\tan \delta$ (figure 2g).
3. The authors switch between different materials without justification. For instance, in Figure 2, data is presented from experiments on PFG-5 to PFG-9 in (b), PFG-6 in (c), PFG-4 in (d) and (e), PFG-1 to PFG-4 in (f) and (g), polymer melts in (h) and some other material with varying molecular weight of the polymer melt in (i) and (j). Why can't these data sets be presented for a consistent set of material(s)? It appears from this style of presentation of data that the authors are cherry picking the best data sets from different experiments.
4. The key material that the authors describe in this manuscript are the PFGs with mixtures of polymers in the melt phase (PFG-b series). The damping response for one such material is shown in figure 3b, along with the contributions from the three components in the melt phase. However, the response of the 102k PBA fluid itself is fairly close to the response of the mixed PFG, and the $\tan \delta$ is always > 0.5 . In this case, what is the justification of using a mixture as the liquid phase?
5. The authors make various unsubstantiated statements throughout the manuscript. For example, the statement "These properties should be ... constituents individually" on page 6-7 doesn't provide any rationale for how or why the hybrid materials are better than the individual components. Similarly, the authors evoke the model from Doi and Edwards on page 9 without providing enough justification for why the model maybe applicable and what are the values of the different parameters they use to arrive at eq. 2 from eq. 1.
6. The logic behind the justification that since the molecular weight of a crosslink strand is larger than the entanglement molecular weight, the entanglements control the maximum elongation in the PFG materials (lines 110-117) is not clear. The authors estimate the molecular weight of a crosslink strand from the maximum elongation strain. Then how can the entanglements control the

maximum elongation? Also, since the polymer chains in the melt can reptate and relax completely (as also shown in figure 2h), wouldn't the measurements of the maximum elongation, if controlled by elongation, be also dependent on the rate of elongation? Relatedly, did the authors observe any dependence of the tensile response on the rate of elongation?

7. The organization and the language of the manuscript can be improved significantly. The authors insist on using sample names PFG-X to identify their materials, but there is not easy way to correlate the names with the materials. Furthermore, too many acronyms are used in the text to be able to follow and differentiate one material from another.

8. The authors should include a discussion on the stability of these PFG materials.

Reviewer #3:

Remarks to the Author:

Authors have proposed that polymer fluid gels have the capability to provide 200 m times higher energy dissipation over a wider range of frequency compared to conventional damping materials. The experiments are performed reasonably in detail and work is certainly of great interest to the scientific community. However, considering that there have been studies where polymer gels have been reported (see Lenhart et al. 10.1088/0964-1726/25/2/025004) to be useful in tunable damping, claiming the work to be fully novel is not justified. Besides there are many issues with the work which I discuss in detail below. In view of these issues (several major) I do not support the publication of this manuscript in a reputed journal as Nature Communications.

Major comments:

- o The Introduction section is poorly written and the motivation does not jump to the reader.
- o First a discussion on the desired properties in a damping material, which is critical to the work, is missing. Second, a detailed discussion on the state of the art on the damping materials is also missing. Authors claim that their PFGs are better than the conventional damping materials, but no proper explanation has been provided to substantiate the claim. What are the conventional damping materials? What are the issues with those? What frequency range they offer and what is the desired range? Such details are entirely missing which makes the motivation weak. Recent articles discussing the use of polymer gels as damping materials (an example is mentioned earlier) have not been properly acknowledged.
- o In a simple melt, polymer chains have physical crosslinks in the form of entanglements. In these systems, few entanglements have been replaced by permanent crosslinks. So, I don't understand the idea of polymer being confined. I don't think polymer can be considered to be confined, unless they deviate from their equilibrium conformation. Is it so?
- o Authors claim that the damping originates from the frictional losses of free polymer chains during their reptation. Based on this claim, damping should always be more for the higher Mw free polymers. Authors also claim that damping occurs for the region where tan delta is maximum, which is not same as having higher frictional losses. Overall, the mechanism and the explanation of damping remains poorly discussed.
- o I don't see any significant advantage of the material being a single component. Miscibility could be a concern, but a combination of polymeric entities which have negative Flory-Huggins interaction parameter can be used to ensure miscibility.
- o Introduction focuses on damping via frictional losses. How is that related to stretchability discussed in results section?

Besides these major issues there are several minor issues. Here, I will just mention a few for authors' benefit:

- o What are PFG 5 and 6? Why aren't they not mentioned in table 1?
- o There are several grammatical issues with the writing and construction of statements is poor.
- o Why have the authors chosen a flexible polymer? Why PBA?
- o What is the state of the art stretchability of other polymer-based materials?
- o What to the authors mean by van der Waals interaction being 'favorable'?
- o Why is the x-axis of figure 2j $Mn^{0.5}$ instead of just Mn? Is the data on Y axis tan (δ) or peak tan(δ) ?

Reviewers' comments:

Reviewer #1 (Remarks to the Author):

This is an interesting, comprehensive, and quite clearly-described study where the authors propose a novel concept for vibration-damping materials based on gels infused with free polymers (in their liquid state), where the gel network chains and the free polymer are based on identical monomers (poly (n-butyl acrylate) (PBA) in the present study). I like the conceptual novelty of tailoring the relaxation characteristics of the free chains together with the gel-network characteristics and the use of multiple free-chains of different molecular weights to “fine-tune” the damping properties of the composite gel. The results are impressive. There are a number of issues, largely of clarification, that the authors should address, as detailed below.

1. Polymer-infused gels have been described in the literature, e.g. Mu et al., *Extreme Mechanics Letters* 38 (2020) 100742; Osaheni et al., *Acta Biomater.* 46, (2016) 245–255.; *Clin. Biomech.* 67 (2019) 15–19. These should be related to.
2. Soft gels have been used for vibration damping. In particular, a commercial vibration damping gel (I believe there may be others), <https://taica.co.jp/gel/en/about/>. This appears to be a silicone gel imbued with low M siloxane oligomers. This should be noted, and also see below in connection with comparisons.
3. I found it made hard reading constantly to refer to the Supplementary Material, especially when referring to different samples. Thus (line 106) PFG-6 and PFG-5 are compared but not clear what they are without chasing up the supplementary. At least the samples should all be in a single table in the main text.
4. I have difficulty understanding the origins of the 50-fold (5000%) elongation (l. 107). The maximal elongation goes like $N^{1/2}$, where N is the number of Kuhn steps in the chain being stretched; but the total number of monomers N monomer either in an entanglement length ($10^3 - 10^4$ Da) or of a cross-link strand (10^5 Da) yield N monomer ≈ 800 at most (given that a monomer molecular weight is ca. 120 Da), and

even assuming each monomer was a Kuhn step (which it isn't! the number of Kuhn steps is much smaller), then $N^{1/2} < \text{ca. } 30$; not 50. I wonder if at 50-fold extension some network links break to give this huge extension? This would be in line with the 250% residual strain (which by the way is quite a large residual strain and points to irreversible chain scission) following the 3000% extension (fig. 2c) of PFG-6. The authors should comment on this.

5. I am not sure I understand the point about reducing the number of entanglements (l. 118 - 121). Is this not a topological constant in the case of cross-linked networks? This point should be clarified.

6. Likewise, please clarify why “the properties of such a hybrid are much better than the properties of its constituents individually” (l. 130).

7. Are the rheological data reversible (in view of the hysteresis in fig. 2c)?

8. Not clear how the PFG can be “soft and brittle” at the same time? (l. 171) Please clarify.

9. Comparison with other vibration damping materials: I am not sure that comparison with bulk rubbers is fully appropriate. I would be happy to see comparison with other gels, especially the existing anti-vibration gel, see comment 2 above.

10. The egg-dropping demonstration is very interesting. But they must specify if the egg was raw (liquid inside the shell) or hard-boiled (solid throughout). If it was liquid then indeed this is a truly remarkable demonstration. If solid, somewhat less so... As above, it would be interesting to compare with what happens to the egg on different damping materials including rubbers and the silicone gel mentioned in 2 above.

11. The comparison of the sand experiments should again be made with a vibration-damping gel if possible rather than with natural rubber (see 2 above).

Overall I feel the idea of regulating the relaxation times of the PFGs by different polymeric fluids and their combinations, and implementing this idea to create vibration-damping materials with remarkable properties, is conceptually-novel, leads to impressive results, and is worthy of publication in Nature Communications. I will be prepared to review a revised version where the above points are addressed.

Minor points:

- a. I think they use the term ‘disperse phase’ (l. 70 and elsewhere) incorrectly. It generally applies to colloidal dispersions, where the disperse phase is the particles, and the continuous phase the surrounding (generally fluid) medium. Why not simply ‘liquid phase’ or ‘fluid phase’?
- b. There is some error in the x-axis of fig. 2j – the $M_n^{1/2}$ values cannot be 3k etc.
- c. At a number of points wad/s appears instead of presumably rad/s (?)
- d. L. 188 – which grey line?

Reviewer #2 (Remarks to the Author):

This manuscript by Huang et al. presents a methodology to create elastomers with tunable damping characteristics. The approach relies on filling a crosslinked polymer network, that serves as the load bearing network, with a polymer melt that aids in energy dissipation. The resulting “filled” elastomers exhibit excellent stretchability and energy dissipation characteristics. In the design aspect, these materials are not different from interpenetrating network elastomers. The innovation, as the authors portray, comes from employing polymers with three molecular weights in the polymer melt component, such that the energy dissipation can be expanded over an extended frequency range. In terms of damping and shock absorbing performance, the demonstrated materials are excellent, although a comparison of the damping performance with contemporary elastomers would have helped the authors convey this point better. Even with the interesting material performance, the publication of this manuscript in Nature Communications cannot be recommended for the following reasons:

1. The authors employ time-temperature superposition (TTS) to arrive at the frequency response of their materials. These frequency sweeps form the basis of a lot

of data presented in figures 2 and 3. However, since the materials are a combination of a network and a polymer melt, the applicability of the TTS approach is questionable. The authors do not report the shift factors for TTS and their behavior for the different materials.

2. The authors also do not discuss the frequency sweeps that they report in figure 2f. Why do G' and G'' plateau at low ω ? What is the dependence of both G' and G'' in the high ω region, and why? Lastly, the frequency sweeps (figure 2f) should be reported for the same frequency range as tan delta (figure 2g).

3. The authors switch between different materials without justification. For instance, in Figure 2, data is presented from experiments on PFG-5 to PFG-9 in (b), PFG-6 in (c), PFG-4 in (d) and (e), PFG-1 to PFG-4 in (f) and (g), polymer melts in (h) and some other material with varying molecular weight of the polymer melt in (i) and (j). Why can't these data sets be presented for a consistent set of material(s)? It appears from this style of presentation of data that the authors are cherry picking the best data sets from different experiments.

4. The key material that the authors describe in this manuscript are the PFGs with mixtures of polymers in the melt phase (PFG-b series). The damping response for one such material is shown in figure 3b, along with the contributions from the three components in the melt phase. However, the response of the 102k PBA fluid itself is fairly close to the response of the mixed PFG, and the tan delta is always > 0.5 . In this case, what is the justification of using a mixture as the liquid phase?

5. The authors make various unsubstantiated statements throughout the manuscript. For example, the statement "These properties should be ... constituents individually" on page 6-7 doesn't provide any rationale for how or why the hybrid materials are better than the individual components. Similarly, the authors evoke the model from Doi and Edwards on page 9 without providing enough justification for why the model maybe applicable and what are the values of the different parameters they use to arrive at eq. 2 from eq. 1.

6. The logic behind the justification that since the molecular weight of a crosslink strand is larger than the entanglement molecular weight, the entanglements control the

maximum elongation in the PFG materials (lines 110-117) is not clear. The authors estimate the molecular weight of a crosslink strand from the maximum elongation strain. Then how can the entanglements control the maximum elongation? Also, since the polymer chains in the melt can reptate and relax completely (as also shown in figure 2h), wouldn't the measurements of the maximum elongation, if controlled by elongation, be also dependent on the rate of elongation? Relatedly, did the authors observe any dependence of the tensile response on the rate of elongation?

7. The organization and the language of the manuscript can be improved significantly. The authors insist on using sample names PFG-X to identify their materials, but there is not easy way to correlate the names with the materials. Furthermore, too many acronyms are used in the text to be able to follow and differentiate one material from another.

8. The authors should include a discussion on the stability of these PFG materials.

Reviewer #3 (Remarks to the Author):

Authors have proposed that polymer fluid gels have the capability to provide 200 times higher energy dissipation over a wider range of frequency compared to conventional damping materials. The experiments are performed reasonably in detail and work is certainly of great interest to the scientific community. However, considering that there have been studies where polymer gels have been reported (see Lenhart et al. 10.1088/0964-1726/25/2/025004) to be useful in tunable damping, claiming the work to be fully novel is not justified. Besides there are many issues with the work which I discuss in detail below. In view of these issues (several major) I do not support the publication of this manuscript in a reputed journal as Nature Communications.

Major comments:

1. The Introduction section is poorly written and the motivation does not jump to the reader.

2. First a discussion on the desired properties in a damping material, which is critical to the work, is missing. Second, a detailed discussion on the state of the art on the damping materials is also missing. Authors claim that their PFGs are better than the conventional damping materials, but no proper explanation has been provided to substantiate the claim. What are the conventional damping materials? What are the issues with those? What frequency range they offer and what is the desired range? Such details are entirely missing which makes the motivation weak. Recent articles discussing the use of polymer gels as damping materials (an example is mentioned earlier) have not been properly acknowledged.

3. In a simple melt, polymer chains have physical crosslinks in the form of entanglements. In these systems, few entanglements have been replaced by permanent crosslinks. So, I don't understand the idea of polymer being confined. I don't think polymer can be considered to be confined, unless they deviate from their equilibrium conformation. Is it so?

4. Authors claim that the damping originates from the frictional losses of free polymer chains during their reptation. Based on this claim, damping should always be more for the higher Mw free polymers. Authors also claim that damping occurs for the region where tan delta is maximum, which is not same as having higher frictional losses. Overall, the mechanism and the explanation of damping remains poorly discussed.

5. I don't see any significant advantage of the material being a single component. Miscibility could be a concern, but a combination of polymeric entities which have negative Flory-Huggins interaction parameter can be used to ensure miscibility.

6. Introduction focuses on damping via frictional losses. How is that related to stretchability discussed in results section?

Besides these major issues there are several minor issues. Here, I will just mention a few for authors' benefit:

a. What are PFG 5 and 6? Why aren't they not mentioned in table 1?

b. There are several grammatical issues with the writing and construction of

statements is poor.

c. Why have the authors chosen a flexible polymer? Why PBA?

d. What is the state of the art stretchability of other polymer-based materials?

e. What do the authors mean by van der Waals interaction being 'favorable'?

f. Why is the x-axis of figure 2j $M_n^{1/2}$ instead of just M_n ? Is the data on Y axis $\tan(\delta)$ or peak $\tan(\delta)$?

Reviewer #1

This is an interesting, comprehensive, and quite clearly-described study where the authors propose a novel concept for vibration-damping materials based on gels infused with free polymers (in their liquid state), where the gel network chains and the free polymer are based on identical monomers (poly (n-butyl acrylate) (PBA) in the present study). I like the conceptual novelty of tailoring the relaxation characteristics of the free chains together with the gel-network characteristics and the use of multiple free-chains of different molecular weights to “fine-tune” the damping properties of the composite gel. The results are impressive. There are a number of issues, largely of clarification, that the authors should address, as detailed below.

Response: We thank the reviewer for the highly recognition and constructive suggestions of our work. In light of the referees’ comments, we have worked hard to revise the manuscript. Revisions in the main text and Supplementary Information (SI) file are highlighted.

1. Polymer-infused gels have been described in the literature, e.g. Mu et al., *Extreme Mechanics Letters* 38 (2020) 100742; Osaheni et al., *Acta Biomater.* 46, (2016) 245–255.; *Clin. Biomech.* 67 (2019) 15–19. These should be related to.

Response: We appreciate the reviewer’s helpful suggestion. In our revised manuscript, we have been cited the related papers. [26] Mu, R., Yang, J., Wang, Y., Wang, Z., Chen, P., Sheng, H. & Suo Z. *Extrem. Mech. Lett.* **38**, 100742 (2020). [27] Osaheni, A. O., Finkelstein, E. B., Mather, P. T. & Blum, M. M. *Acta Biomater.* **46**, 245-255 (2016). [28] Sismondoa, R. A., Wenera, F. W., Ordwaya, N. R., Osahenib, A. O., Blumb, M. M. & Scuderi M. G. *Clin. Biomech.* **67**, 15-19 (2019). These literatures describe the preparation of hydrogels with low friction by infusing a small amount of hydrophilic polymer that served as a boundary lubricant and promoted a reduction in friction through hydration lubrication. In our manuscript, we have also mentioned that the combination of solvent molecules and polymer chains greatly reduces the internal frictions of polymer materials (L. 70–74).

2. Soft gels have been used for vibration damping. In particular, a commercial vibration damping gel (I believe there may be others), <https://taica.co.jp/gel/en/about/>. This appears to be a silicone gel imbued with low M siloxane oligomers. This should be noted, and also see below in connection with comparisons.

Response: We appreciate the reviewer’s helpful suggestion. Because the formulation of this damping gel is not known by public, we do not know if it contains low M siloxane oligomers. We compared the damping property of the PFGs and the commercial vibration damping gel developed by Taica Company (Japanese). According to the dynamic mechanical data provided by Taica company, the product with the best damping performance exhibits a maximum damping coefficient of about 0.8 and a narrow frequency range ($10^1 \sim 10^4$ Hz) of $\tan\delta > 0.5$, which is much narrower than that of our work ($10^{-2} \sim 10^8$ Hz), as shown below. Meanwhile, the loss factor of the damping gel is low near room temperature or low-frequency stage.

Figure R1. Dynamic mechanical master curves of the damping gel with best damping property in all products. (Quote: https://taica.co.jp/gel/en/product/shock_absorption/theta_sheet.html)

Figure S14. Frequency dependence of storage moduli (G'), loss moduli (G''), and loss factor ($\tan\delta$) for PFG-b1 were obtained at 25°C with the shear strain of 0.5%.

3. I found it made hard reading constantly to refer to the Supplementary Material, especially when referring to different samples. Thus (line 106) PFG-6 and PFG-5 are compared but not clear what they are without chasing up the supplementary. At least the samples should all be in a single table in the main text.

Response: We appreciate the reviewer’s helpful suggestion. In our revised manuscript, we have added Table 1 including all sample parameters.

Table 1. Molecular parameters and mechanical characteristics of PFGs.

Samples	$\Phi_{\text{PBA fluid}}^a$ (%)	Φ_c^b (%)	$M_n(\text{PBA fluid})$ (kg/mol)	Plateau modulus G_p^c (kPa)	Compressive modulus E^d (kPa)
PFG _(3%)	0	3	/	132.0	453.6
PFG _(3%,20%-35k)	20	3	35.1	54.9	276.2
PFG _(3%,40%-35k)	40	3	35.1	14.4	94.2
PFG _(3%,60%-35k)	60	3	35.1	4.6	18.1
PFG _(0.1%)	0	0.1	5.2	/	/
PFG _(0.1%,40%-5k)	40	0.1	5.2	/	/

4. I have difficulty understanding the origins of the 50-fold (5000%) elongation (l. 107). The maximal elongation goes like $N^{1/2}$, where N is the number of Kuhn steps in the chain being stretched; but the total number of monomers N monomer either in an entanglement length ($10^3 - 10^4$ Da) or of a cross-link strand (10^5 Da) yield N monomer ≈ 800 at most (given that a monomer molecular weight is ca. 120 Da), and even assuming each monomer was a Kuhn step (which it isn’t! the number of Kuhn steps is much smaller), then $N^{1/2} < \text{ca. } 30$; not 50. I wonder if at 50-fold extension some network links break to give this huge extension? This would be in line with the 250% residual strain (which by the way is quite a large residual strain and points to irreversible chain scission) following the 3000% extension (fig. 2c) of PFG-6. The

authors should comment on this.

Response: We appreciate and agree with the reviewer's comments. There are two possible explanations for the 50-fold (5000%) elongation of the PFG. The first one is what the reviewer suggested: at 50-fold extension some network links break to give this huge extension. This is in line with the 250% residual strain, which points to irreversible chain scission following the 3000% extension (Fig. 2c) of PFG_(0.1%, 40%-5k). The other point is that the regular network model is assumed in the theoretical estimation of the maximum elongation, while the PFG network is irregular. Polymer networks are formed via the free radical polymerization process. Consequently, they possess topological defects, which can improve the tensile rate to some extent.

5. I am not sure I understand the point about reducing the number of entanglements (l. 118 - 121). Is this not a topological constant in the case of cross-linked networks? This point should be clarified.

Response: We appreciate the reviewer's helpful question. For an established polymer network, the number of entanglements is indeed a constant, which will not change with the introduction of the fluid or solvent. In our case, the PFG is formed by free radical polymerization in the prepolymer containing monomers, cross-linking agents and a large number of unentangled polymer fluids. In the crosslinking process, the propagating chains are filled with polymer fluids to prevent them from forming entanglements. This situation is similar to hydrogels. The entanglements of the networks are greatly reduced due to the large amount of solvent molecules in the polymerization process.

6. Likewise, please clarify why "the properties of such a hybrid are much better than the properties of its constituents individually" (l. 130).

Response: We appreciate the reviewer's helpful suggestion. The PFGs are new structural elastomers composed of polymer fluid and network with single component, which combine the elasticity of a network with the viscosity of a fluid. According to the uniaxial static mechanical data, the PFGs exhibit better stretchability and

compressibility than that of the pure PBA network (Fig. 2b and Supplementary Fig. 4). Therefore, this sentence expresses that the stretchability and compressibility of such a hybrid are better than that of its polymer network or fluid individually (L. 132).

Figure 2b. Tensile stress-strain curves of the PFGs.

Figure S4. Compressive stress-strain curves of the PFG_(3%, 60%-35k) and the PFG_(3%).

7. Are the rheological data reversible (in view of the hysteresis in fig. 2c)?

Response: We appreciate the reviewer’s helpful question. The hysteresis in Fig. 2c is caused by irreversible chain scission of the polymer network under deformation beyond its linear viscoelastic region (3,000%). In response to this question, the characterization of linear viscoelastic region for the PFG has been added to the revised Supplementary Material (Supplementary Fig. 5). As shown in Supplementary Fig. 5, the PFG is in the linear viscoelastic region within 10% of the strain. The rheological data are measured at 0.5% strain, so they are reversible. We have also

added the explanation and experimental methods in the revised manuscript (L. 135–137 and L. 297–299).

Figure S5. The shear strain (γ) of storage moduli (G') and loss moduli (G'') for $\text{PFG}_{(3\%, 60\%-35k)}$ were obtained at the frequency of 10 rad/s and temperature of 25 °C.

8. Not clear how the PFG can be “soft and brittle” at the same time? (l. 171) Please clarify.

Response: We appreciate the reviewer’s question. Due to the low M_n of the infused PBA fluids, they are too short to be entangled and their whole chain relaxation is very fast. As shown in Fig. 2j, the PFG exhibits low energy dissipation (low $\tan\delta$) and modulus (low G'). In this case, PFGs are soft and brittle, and their mechanical properties are similar to that of the liquid gels, like fragile jellies (L. 180).

9. Comparison with other vibration damping materials: I am not sure that comparison with bulk rubbers is fully appropriate. I would be happy to see comparison with other gels, especially the existing anti-vibration gel, see comment 2 above.

Response: We appreciate the reviewer’s helpful suggestion. Below, we systematically compared PFG with the damping gel.

Damping property: the damping properties have been compared in response 2. The damping gel exhibits a narrow frequency range of $\tan\delta > 0.5$ ($10^1 \sim 10^4$ Hz),

while PFG has a wide range ($10^{-2} \sim 10^8$ Hz).

Energy dissipation: the energy dissipation (ΔW) of the PFG_(3%, 60%-35k) is over 3 times higher than that of the damping gel under the same periodic sinusoidal alternating stress (15 kPa), as shown in Figure R2.

Figure R2. Large amplitude oscillatory shear tests of the PFG_(3%, 60%-35k) and the damping gel.

Shock absorption: an impulse force of 35 N was applied to the damping materials by a drop weight. As shown in Figure R3, PFG_(3%, 60%-35k) can reduce the impact force up to 85%, while the reduction for the damping gel is 62%.

Figure R3. Drop weight impact tests on the PFG_(3%, 60%-35k) and the damping gel.

Vibration absorption: the vibration absorption capacity was measured by shaking

table demonstrative experiments. The amplitude of the vibration signal with at 1,000 Hz applied was dissipated by 48% for the damping gel, whereas the reduction for PFG_(3%, 60%-35k) is 90% (Figure R4).

Figure R4. Shaking table demonstrative experiments of the PFG_(3%, 60%-35k) and the damping gel.

In conclusion, the damping performance of the PFGs is superior in comparison to the commercial damping gel.

10. The egg-dropping demonstration is very interesting. But they must specify if the egg was raw (liquid inside the shell) or hard-boiled (solid throughout). If it was liquid then indeed this is a truly remarkable demonstration. If solid, somewhat less so... As above, it would be interesting to compare with what happens to the egg on different damping materials including rubbers and the silicone gel mentioned in 2 above.

Response: We appreciate and agree with the reviewer's comments. Following the reviewer's suggestion, we verified that the egg is raw (Supplementary Movie 4 and Supplementary Fig. 17) and changed the phase 'egg' to 'raw egg' in the revised manuscript (L. 240). Moreover, we demonstrated eggs were dropped from 2 m height onto the 5 mm-thick natural rubber and damping gel. The results indicate that eggs were all severely cracked (Supplementary Movie 5 and R1). We have added these results in the revised manuscript (L. 242).

Figure S17. The egg was dropped from 2 m height onto a 5 mm-thick PFG_(1%, 60%-35k) pad and remained unbroken.

11. The comparison of the sand experiments should again be made with a vibration-damping gel if possible rather than with natural rubber (see 2 above).

Response: We appreciate the reviewer's helpful comments. The damping materials with 5mm height were placed under the petri dishes containing sands and put on an ultrasonic generator. The results show that the sands atop the damping gel spacer beat, whereas the sands atop the PFG_(3%, 60%-35k) spacer stay still (Movie R2).

Overall I feel the idea of regulating the relaxation times of the PFGs by different polymeric fluids and their combinations, and implementing this idea to create vibration-damping materials with remarkable properties, is conceptually-novel, leads to impressive results, and is worthy of publication in Nature Communications. I will be prepared to review a revised version where the above points are addressed.

Response: We would like to again express our appreciation to the referees for her/his helpful comments and constructive suggestions. We realize that the reviewer spent a considerable amount of time carefully reading the manuscript and formulating a strategy of action for us to make this paper better. We are grateful for these.

Minor points:

a. I think they use the term ‘disperse phase’ (l. 70 and elsewhere) incorrectly. It generally applies to colloidal dispersions, where the disperse phase is the particles, and the continuous phase the surrounding (generally fluid) medium. Why not simply ‘liquid phase’ or ‘fluid phase’?

Response: We thank the reviewer for pointing this out. According to the reviewer’s helpful suggestion, all the terms ‘disperse phase’ have been replaced by ‘fluid phase’ in the revised manuscript (L. 71 and L. 74).

b. There is some error in the x-axis of fig. 2j – the $M_n^{1/2}$ values cannot be 3k etc.

Response: We thank the reviewer for pointing this out. We have corrected some errors in the x-axis of Fig. 2j in the revised manuscript.

c. At a number of points wad/s appears instead of presumably rad/s (?)

Response: We thank the reviewer for pointing this out. All the terms of ‘wad/s’ have been replaced by ‘rad/s’ in the revised manuscript (L. 179, 182 and 214).

d. L. 188 – which grey line?

Response: We appreciate the reviewer’s comments. The grey line is in the Fig. 3a and has been marked in the revised manuscript (L. 198).

Reviewer #2 (Remarks to the Author):

This manuscript by Huang et al. presents a methodology to create elastomers with tunable damping characteristics. The approach relies on filling a crosslinked polymer network, that serves as the load bearing network, with a polymer melt that aids in energy dissipation. The resulting “filled” elastomers exhibit excellent stretchability and energy dissipation characteristics.

Response: We thank the reviewer for the positive evaluation of our manuscript.

In the design aspect, these materials are not different from interpenetrating network elastomers.

Response: We appreciate the reviewer’s comments. We would like to clarify that our work is entirely different from interpenetrating network elastomers. The comparisons of our works and the interpenetrating polymer network (IPN) elastomers are as follows:

Table R1. The comparison of the PFGs and the IPN elastomers.

	Component	topological structure	Damping mechanism	$\tan\delta > 0.5$ (°C width)	Damping controllability
IPN	Two or more	Interpenetrating network	Chain segment relaxation	~ 60	No
PFG	One	network + fluid	Whole chain relaxation	~ 170	Yes

The IPN elastomers: The IPN elastomers are formed of two or more polymer components, each of which is a crosslinked three-dimensional network, and one of which is formed in the presence of the other. The polymer networks are physically entangled with, but not covalently bonded to, each other. These networks do not dissolve in solvent or flow when heated (*L. H. Sperling, Interpenetrating Polymer Networks and Related Materials, Plenum Press, New York, 1981*).

The IPN network elastomers (eg. polyurethane/epoxy) are one of the typical

classical damping materials because of their broad glass transition regions. This is associated with the dynamic heterogeneity of chain segment relaxation often obtained for various polymer components, as mentioned in the introduction section (L. 42). However, this strategy cannot significantly broaden the effective damping region due to the limitation of inherently narrow glass transition regions for general polymer materials, with a frequency range normally spanning 10^2 Hz. In addition, their damping property cannot be precisely controlled.

The PFGs: the PFGs are a single-component hybrid by infusing the polymer fluids into the elastic networks. The polymer fluids component does not form a permanent network.

The PFGs utilize the internal friction arising from the whole chain relaxation of the confined fluids in the network to realize energy dissipation. By precisely regulating the relaxation time of polymer fluids, PFGs can present optimal energy dissipating at a desired frequency. Furthermore, the general idea is exploited for the design of a high energy-dissipation property over a wide frequency range through infusing several polymer fluids with significantly different chain lengths into the matrix.

Overall, the PFGs are completely different from the IPN network elastomers in topological structures and damping mechanisms.

The innovation, as the authors portray, comes from employing polymers with three molecular weights in the polymer melt component, such that the energy dissipation can be expanded over an extended frequency range. In terms of damping and shock absorbing performance, the demonstrated materials are excellent, although a comparison of the damping performance with contemporary elastomers would have helped the authors convey this point better.

Response: We thank the reviewer for the positive evaluation of our manuscript.

Even with the interesting material performance, the publication of this manuscript in Nature Communications cannot be recommended for the following reasons:

Response: We appreciate the reviewer's comments. According to the reviewer's

comments, we have explained and verified the following reasons in detail. Revisions in the main text and Supplementary Information (SI) file are highlighted.

1. The authors employ time-temperature superposition (TTS) to arrive at the frequency response of their materials. These frequency sweeps form the basis of a lot of data presented in figures 2 and 3. However, since the materials are a combination of a network and a polymer melt, the applicability of the TTS approach is questionable. The authors do not report the shift factors for TTS and their behavior for the different materials.

Response: We thank the reviewer for this suggestion. In principle, the immiscible blends may cause failure of the TTS principle. The reason is that the components retain their own temperature sensitivity in the blend due to heterogeneities on a very small scale, while the local dynamics may be influenced by each component's surroundings (van Gorp, M.; Palmen, J. *Rheol. Bull.* **67**, 5–8 (1998)). For our works (PFGs), the chemical composition of both the network and the fluid are PBA, forming the strong coupling that leads to one single temperature dependence. Obviously, when the temperature sensitivities of both components are comparable, TTS will hold.

Following the reviewer's suggestion, we utilized Williams-Landel-Ferry (WLF) equation (Williams, M. L., Landel, R. F. & Ferry J. D. *J. Am. Chem. Soc.* **77**, 3701–3707 (1955)) and van Gorp-Palmen plots diagram (van Gorp, M.; Palmen, J. *Rheol. Bull.* **67**, 5–8 (1998)) to further verify that PFGs obey TTS. Meanwhile, we also reported the shift factors for TTS.

WLF equation: G' values were measured over a 10^1 – 10^2 Hz frequency range at different temperatures. Supplementary Fig. 6 presents application of the TTS for data of PFG_(3%,60%-35k). Supplementary Fig. 6a shows the behavior of G' at different temperatures (-35–85 °C, $\Delta T = 10.0$ °C, where ΔT describes the temperature interval) as a function of frequency and G' decreases with increasing temperature and decreasing frequency. Supplementary Fig. 6b shows the master curve obtained by shifting the data in Supplementary Fig. 6a horizontally along the frequency axis with a reference temperature (T_{ref}) of 25°C.

Figure S6. Application of the time–temperature superposition principle. (a) G' data of PFG_(3%, 60%-35k) over a 10^{-1} – 10^1 Hz frequency range and a -35 – 85 °C temperature range. (b) The time–temperature master curve made from experimental data in (a) with $T_{ref} = 25$ °C.

From the data in the Supplementary Fig. 6b, we can calculate the temperature-dependent shift factors (α_t) (Supplementary Fig. 7a, black \blacklozenge). We chose α_t at 55°C and α_t at -5°C to calculate the empirical fitting constants (C_1 , C_2) of WLF equation (1) (Supplementary Fig. 7b). The red line is a fit by equation (1) in the Supplementary Fig. 7a. The fitted curve is ideally matched with the experimental data α_t (black \blacklozenge). The results indicate that PFGs obey TTS.

Figure S7. Williams-Landel-Ferry (WLF) equation. (a) The temperature-dependent shift factors (α_t) of the PFG_(3%, 60%-35k) as a function of the temperature (black \blacklozenge) and

the fitted curve of WLF equation with WLF parameters $C_1 = 6.51$ and $C_2 = 156$ °C for $T_{\text{ref}} = 25$ °C (red line). (b) Calculation of the WLF parameters C_1 and C_2 for $T_{\text{ref}} = 25$ °C.

Van Gulp-Palmen plots: The van Gulp-Palmen (vGP) plot is classically used to determine if TTS is appropriate for polymers. The loss angle δ ($\delta = \text{atan}(G'' / G')$) is used to determine if TTS works because δ is invariant to temperature changes, owing to the density differences are cancelled out by the ratio of G' to G'' . The loss angle δ versus the complex modulus (G^*) was plotted. This way of plotting eliminates the effect of shifting along the frequency axis, and yields temperature independent curves. When the plots are continuous, it indicates that TTS holds. Therefore, successful application of TTS for the PFG_(3%, 60%-35k) can conveniently be read from Supplementary Fig.S8.

Figure S8. Van Gulp-Palmen plots of PFG_(3%, 60%-35k).

We have added these contents in the revised manuscript (L. 137–140 and 304-318)

2. The authors also do not discuss the frequency sweeps that they report in figure 2f. Why do G' and G'' plateau at low ω ? What is the dependence of both G' and G'' in the high ω region, and why? Lastly, the frequency sweeps (figure 2f) should be reported

for the same frequency range as tan delta (figure 2g).

Response: We thank the reviewer for this suggestion.

What is the dependence of both G' and G'' in the high ω region, and why?

In the high ω region: for linear polymers and crosslinked polymers, in the high ω (low T) region (glass transition region), the short-range diffusional motions of polymer chains begin. With decreasing ω , the thermal energy becomes roughly comparable to the potential energy barriers to segment rotation and translation. Segments are free to “jump” from one lattice site to another; the brittle glass becomes a resilient leather. This region is accompanied by a sharp decrease in the modulus (*Introduction to Polymer Viscoelasticity*, Wiley, New York, 1983).

Why do G' and G'' plateau at low ω ?

In the low ω region: for the crosslinked polymers, the crosslinks consisting of primary chemical bonds remain intact, preventing the chains from translating relative to one another. Thus, although the modulus changes slightly with temperature in the rubbery plateau region of the crosslinked polymers, the changes are small compared to those exhibited during the glass transition. The situation is quite different for a linear polymer. Local chain interactions are no longer of sufficiently high energy to prevent molecular flow. The molecules will slip by one another, and the polymer sample will exhibit a correspondingly low modulus (*Introduction to Polymer Viscoelasticity*, Wiley, New York, 1983). In this case, for the PFGs, the polymer fluids (linear polymers) in the network are in viscous flow state, and their modulus is very low. Therefore, the PFGs' modulus is mainly contributed by the network, which is also a platform region similar to the crosslinked polymers. We have added the discussion in the revised manuscript (L. 145-147).

Lastly, the frequency sweeps (figure 2f) should be reported for the same frequency range as tan delta (figure 2g).

The plateau modulus (G_p) of PFGs appears in the low frequency region ($\omega \rightarrow 0$

rad/s) (Fig. 2f), while the whole chain motion of polymer fluids in the PFGs appears in the intermediate frequency ($\omega \sim 158$ rad/s) (Fig. 2g). Therefore, we plot the plateau modulus and the energy dissipation in their representative ranges where the most interesting features appear. According to the reviewer's comments, we added the modulus with the same frequency range as the tan delta (Figure. R5).

Figure R5. Frequency dependence of storage modulus (G') and loss modulus (G'') for PFGs. The master curves were shifted horizontally by the indicating factors for clarity.

3. The authors switch between different materials without justification. For instance, in Figure 2, data is presented from experiments on PFG-5 to PFG-9 in (b), PFG-6 in (c), PFG-4 in (d) and (e), PFG-1 to PFG-4 in (f) and (g), polymer melts in (h) and some other material with varying molecular weight of the polymer melt in (i) and (j). Why can't these data sets be presented for a consistent set of material(s)? It appears from this style of presentation of data that the authors are cherry picking the best data sets from different experiments.

Response: We thank the reviewers for this comment. The purpose of Figure 2 is to show the precise controllability of the mechanical properties of the PFGs. The desired mechanical properties can be achieved by changing the structural parameters of the material.

PFG-5 to PFG-9 describe that excellent tensile properties can be achieved by changing the crosslinking degree and polymer fluid content (Fig. 2b). Among them,

PFG-6 shows the best tensile property and cyclic tensile property (Fig. 2b, c).

PFG-1 to PFG-4 validate that the high damping performance of PFGs can be achieved by adjusting the polymer fluid content in the network. As content of polymer fluids increases, the plateau modulus (G_p) of PFGs decreases from 132 kPa to 4.6 kPa (Fig. 2f), and the peak of $\tan\delta$ for PFGs becomes more evident in the rubbery region of the network matrix (Fig. 2g). It arises from the whole chain reptation of polymer fluids in the polymer network. Figure 2d and 2e show the robust fatigue resistance of the PFG-4 with the best damping performance in PFG-1 to PFG-4, which further verifies that the inclusion of polymer fluids into a polymer network can effectively dissipate the external applied stress.

Figure 2h, 2i and 2j show that the PFGs can present optimal energy dissipating and mechanical properties at a desired frequency through precisely regulating the relaxation time of polymer fluids by varying their chain length (molecular weight).

4. The key material that the authors describe in this manuscript are the PFGs with mixtures of polymers in the melt phase (PFG-b series). The damping response for one such material is shown in figure 3b, along with the contributions from the three components in the melt phase. However, the response of the 102k PBA fluid itself is fairly close to the response of the mixed PFG, and the tan delta is always > 0.5 . In this case, what is the justification of using a mixture as the liquid phase?

Response: We appreciate the reviewer's helpful comments. The justifications of using a mixture as the liquid phase are as follows:

High damping frequency range: We need to explain that the high damping ($\tan\delta > 0.5$) frequency range ($10^{-2} \sim 10^8$ Hz) of the PFG-b1 is slightly wider than that of the PFG with the 102k PBA fluid. From the figure 3a, we can increase the damping performance at lower frequencies ($< 10^{-2}$ Hz) by filling in fluids with higher molecular weights (> 200 kDa). However, there is hardly any source of vibration below 10^{-2} Hz in life, except for some nuclear industries and natural disasters. According to practical applications, we do not blindly expand the range of high damping frequency.

Another reason is that the relaxation region of the whole chain is close to the glass transition region of the network in the PFG with the 102k PBA fluid, which makes the $\tan\delta$ keep a high value. In the PFGs with lower T_g polymers (eg. PDMS), these two regions should be clearly separated, as shown in the figure 1b(ii). In this case, the high damping frequency range of the PFGs with mixtures of fluids is certainly wider than that of the PFGs with a single fluid.

Stability of damping performance: as shown in the figure 3b, the $\tan\delta$ curve of the PFG-b1 clearly displays the relaxation behavior of each PBA fluid component. Compared to the PFG with a single polymer fluid (eg. 102k PBA fluid), the $\tan\delta$ curve of PFG-b1 displays a more stable trend rather than a pronounced peak, owing to the gradual relaxation of various PBA fluids.

Controllability of damping performance: unlike the PFGs with a single PBA fluid, the PFG-bs contain various M_n PBA fluids. The PFG-bs can achieve more possibilities for the damping performance by regulating the PBA fluid ratio.

5. The authors make various unsubstantiated statements throughout the manuscript. For example, the statement “These properties should be ... constituents individually” on page 6-7 doesn’t provide any rationale for how or why the hybrid materials are better than the individual components. Similarly, the authors evoke the model from Doi and Edwards on page 9 without providing enough justification for why the model maybe applicable and what are the values of the different parameters they use to arrive at eq. 2 from eq. 1.

Response: We appreciate the reviewer’s helpful suggestion. The PFGs are new structural elastomers composed of polymer fluid and network with single component, which combine the elasticity of a network with the viscosity of a fluid. According to the uniaxial static mechanical data, the PFGs exhibit better stretchability and compressibility than that of the pure PBA network (Fig. 2b and Supplementary Fig. 4). Therefore, this sentence expresses that the stretchability and compressibility of such a hybrid are better than that of its polymer network or fluid individually (L. 132).

Figure 2b. Tensile stress-strain curves of the PFGs.

Figure S4. Compressive stress-strain curves of the PFG_(3%, 60%-35k) and the PFG_(3%).

Doi and Edwards model: We interpreted our experimental results using the model in Chapter 6 of Doi and Edwards (*The theory of polymer dynamics*, Clarendon Press, Oxford, 1986). This model discussed the dynamics of polymer chains in a fixed network. This is precisely the experimental setting in our manuscript, where the polymer fluids are the polymer component, and the crosslinked networks provide the fixed network. Incorporating the correction due to the contour length fluctuation, the relaxation time is given by

$$\tau = \frac{b^2 \zeta_0 M_n^3}{M_x M_0^2 \pi^2 k_B T} \left[1 - 1.3 \left(\frac{M_x}{M_n} \right)^{0.5} \right]^2 = C M_n^3 \left[1 - 1.3 \left(\frac{M_x}{M_n} \right)^{0.5} \right]^2$$

where b is the Kuhn length, and ζ_0 and M_0 are the frictional coefficient and molecular weight of the monomer, respectively. M_n is the molecular weight of the polymer fluids and $M_x = 4.2$ kDa is the average molecular weight of cross-linking strands in the

network. When we plot the relaxation time as a function of with M_n , the fitting parameter $C = \frac{b^2 \zeta_0}{M_e M_0^2 \pi^2 k_B T} \approx 10^{-4.96}$ is obtained. Here we have found a typo in the fitting parameter in the previous manuscript. We have reported the corrected value (Fig. 3a), and added the explanation for the applicability of Doi and Edwards model in the revised manuscript (L. 197–200).

Figure R6. The relaxation time (τ) of polymer fluids as a function of the molecular weight (M_n).

To check the validity of the fitting parameter, we also plot the relaxation time of pure PBA melts as a function of M_n . For this case, the network is replaced by the self-entanglement between polymer chains. The relaxation time is modified as

$$\tau_{melt} = \frac{b^2 \zeta_0 M_n^3}{M_e M_0^2 \pi^2 k_B T} \left[1 - 1.3 \left(\frac{M_e}{M_n} \right)^{0.5} \right]^2 = \left(C \frac{M_n}{M_e} \right) M_n^3 \left[1 - 1.3 \left(\frac{M_e}{M_n} \right)^{0.5} \right]^2$$

where $M_e = 20$ kDa is the entanglement molecular weight for PBA. One can see that the experimental data approaches the predicted line in the limit of large molecular weight. Using the fitting parameter for the PFG, we obtained good agreement of the relaxation times for both the PFG and the melt. This gives us confidence in the validity of the model.

6. The logic behind the justification that since the molecular weight of a crosslink strand is larger than the entanglement molecular weight, the entanglements control the maximum elongation in the PFG materials (lines 110-117) is not clear. The authors

estimate the molecular weight of a crosslink strand from the maximum elongation strain. Then how can the entanglements control the maximum elongation? Also, since the polymer chains in the melt can reptate and relax completely (as also shown in figure 2h), wouldn't the measurements of the maximum elongation, if controlled by elongation, be also dependent on the rate of elongation? Relatedly, did the authors observe any dependence of the tensile response on the rate of elongation?

Response: We thank the reviewer for this comment. 'Control' in the article refers to the meaning of 'limit'. For conventional linear-chain polymer networks, the maximum elongation (λ_{\max}) is scaled as $\lambda_{\max} \propto N_x^{1/2}$, where N_x is the degree of polymerization of segments between the chemical cross-linked points (N_c) or entanglement points (N_e) depending on their magnitude (Vatankhahvarnosfaderani, M. *et al. Nature* **549**, 497–501 (2017)). Typical linear polymers possess $M_e \cong 10^3 \sim 10^4$ g mol⁻¹. For the N_c , we can regulate the cross-linking density to make N_c much larger than N_e ($N_c \gg N_e$). Thus, the entanglement controls the maximum elongation at break. This is similar to the barrel principle, which is that the capacity of the barrel depends on the length of the shortest board. To increase the elongation at break, we must reduce the total number of entanglements. When entanglements are largely diluted out by unentangled PBA fluids, the maximum elongation is controlled by the molecular weight of a crosslink strand.

The tensile test sample (PFG_(0.1%, 40%-5k)) was infused with unentangled PBA fluids ($M_n = 5k$), where the whole chain relaxation of PBA fluids has already occurred at high frequencies ($\omega > 10^4$ rad/s) (Fig. 2i). Therefore, the maximum elongation exhibits tensile rate dependency only if the tensile rate is very high. Due to the limited stretching rate of the tensile machine, we did not observe dependence of the tensile response on the rate of elongation.

7. The organization and the language of the manuscript can be improved significantly. The authors insist on using sample names PFG-X to identify their materials, but there is not easy way to correlate the names with the materials. Furthermore, too many acronyms are used in the text to be able to follow and differentiate one material from

another.

Response: We appreciate the reviewer's helpful suggestions. We have renamed the samples in the revised manuscript. We changed the PFG-X to the $\text{PFG}_{(\phi_c, \phi_{\text{PBA fluid}}-M_n)}$, such as $\text{PFG}_{(3\%, 60\%-35k)}$.

8. The authors should include a discussion on the stability of these PFG materials.

Response: We thank the reviewer for this comment. In our paper, we have discussed the stability of these PFG materials. The PFGs demonstrate robust fatigue resistance (Fig. 2e) and high stability (Supplementary Fig. 3). $\text{PFG}_{(3\%, 60\%-35k)}$ still holds original compressive strength after 1,000 compressive cycles at 60% strain. The mass of the $\text{PFG}_{(3\%, 60\%-35k)}$ remains almost unchanged at room temperature for 900h.

Figure 2e. Cyclic stress-strain curves of the $\text{PFG}_{(3\%, 60\%-35k)}$ compressed to 60% strain and released for 1000 cycles.

Figure S3. The mass loss rate of the $\text{PFG}_{(3\%, 60\%-35k)}$ and the PBA organogel after 900h at room temperature.

Reviewer #3 (Remarks to the Author):

Authors have proposed that polymer fluid gels have the capability to provide 200 times higher energy dissipation over a wider range of frequency compared to conventional damping materials. The experiments are performed reasonably in detail and work is certainly of great interest to the scientific community.

Response: We thank the reviewer for the positive evaluation of our manuscript.

However, considering that there have been studies where polymer gels have been reported (see Lenhart et al. 10.1088/0964-1726/25/2/025004) to be useful in tunable damping, claiming the work to be fully novel is not justified.

Response: We appreciate the reviewer's comments. We would like to clarify that our work is entirely different from the polymer gels reported by Lenhart et al (we named them 'L-gels'). The comparisons of our works and the L-gels are as follows:

Topological structure: the L-gels consist of polymer networks and mineral oils (organic small molecules), which belong to the traditional liquid gels. These materials have indeed been widely reported, such as hydrogels and organogels. Our works (PFGs) consist of networks and polymer fluids. Thus, the topological structure of PFGs is fundamentally different from that of L-gels.

Damping mechanism: in the L-gels, the physical cross-linked chains (thermally reversible) significantly outnumber the covalent cross-links (permanent). Upon application of heat, the physical cross-links dissociate resulting in an increased number of dangling chain ends, where the dangling chain ends promote energy dissipation to provide enhanced mechanical damping. This mechanism of utilizing dangling chains to realize energy dissipation is also summarized in the introduction section (L. 43).

Our works (PFGs) utilize the internal friction arising from the whole chain relaxation of the confined fluid in the network to realize energy dissipation.

Damping properties: At 25 °C, the loss factor ($\tan\delta = G'' / G' < 0.1$) of L-gels damping coefficient of the polymer is very low, which is similar to that of the liquid

gels. Only when the temperature is above 60 °C, the L-gels dissociate show excellent damping performance, due to the dissociation of their physical cross-links. These gels focus on mechanical switching according to the local environment rather than the tunable damping performance.

The PFGs' energy-dissipation property can be precisely tailored at desired frequencies through adjusting the relaxation time of polymer fluids by varying chain length. Moreover, the PFGs exhibit high energy-dissipation performance (loss factor > 0.5) with a broad frequency ($10^{-2} \sim 10^8$ Hz)/temperature (-50 ~ 120 °C) by infusing several polymer fluids with significantly different chain length.

Besides there are many issues with the work which I discuss in detail below. In view of these issues (several major) I do not support the publication of this manuscript in a reputed journal as Nature Communications.

Response: We appreciate the reviewer's very helpful comments. In light of the referees' comments, we have explained and modified the following reasons in detail, in the revised manuscript. Revisions in the main text and Supplementary Information (SI) file are highlighted.

Major comments:

1. The Introduction section is poorly written and the motivation does not jump to the reader. First a discussion on the desired properties in a damping material, which is critical to the work, is missing. Second, a detailed discussion on the state of the art on the damping materials is also missing. Authors claim that their PFGs are better than the conventional damping materials, but no proper explanation has been provided to substantiate the claim. What are the conventional damping materials? What are the issues with those? What frequency range they offer and what is the desired range? Such details are entirely missing which makes the motivation weak. Recent articles discussing the use of polymer gels as damping materials (an example is mentioned earlier) have not been properly acknowledged.

Response: We thank the reviewer for this comment.

First a discussion on the desired properties in a damping material, which is critical to the work, is missing. Second, a detailed discussion on the state of the art on the damping materials is also missing.

In the introduction section, we have mentioned the desired properties (L. 34–35) and the critical works (the state of the art on the damping materials) (L. 40–46). The desired properties of the damping materials are high energy dissipation over a broad frequency/temperature range. The critical works of the damping materials is to tune dynamic heterogeneity of polymer chains to broaden the frequency range, such as adding nanofillers, blending polymers, polymer/organic molecules hybrids, copolymerization, gradient polymers, interpenetrating polymer networks, and polymers with dangling chains. However, all these strategies cannot significantly broaden the effective damping region due to the limitation of inherently narrow glass transition regions for general polymer materials, with a frequency range normally spanning 10^2 Hz.

What are the conventional damping materials? What are the issues with those?

The conventional damping materials are mainly composed of amorphous polymers with unique viscoelastic property because of the high energy dissipation induced by strong internal friction of chain relaxation in the glass transition region. We have added this content in the revised manuscript (L. 35–38). The issues with them are that the effective damping region is narrow and a large drop in modulus about three or four orders of magnitude is unavoidable in a small temperature variation (L. 43–47).

Authors claim that their PFGs are better than the conventional damping materials, but no proper explanation has been provided to substantiate the claim.

We have provided proper explanation to substantiate the claim. The PFGs' energy-dissipation property can be precisely tailored at desired frequencies. The PFGs exhibit high energy-dissipation performance (loss factor > 0.5) over a broad frequency range ($10^{-2} \sim 10^8$ Hz), which exceeds typical state-of-the-art damping materials (L.

58-60). Moreover, the PFGs exhibit a considerably higher shock and vibration absorption than that of the conventional damping materials (L. 60–63). These properties are verified in the Results section.

What frequency range they offer and what is the desired range?

We provide the effective frequency range of $10^{-2} \sim 10^8$ Hz. The desired frequency range is the frequency of the source of vibration in the practical application.

Recent articles discussing the use of polymer gels as damping materials (an example is mentioned earlier) have not been properly acknowledged.

We appreciate the reviewer's helpful suggestion. In our revised manuscript, we have been cited the related paper. [22]. Mrozek, R. A., Berg, M. C., Gold, C. S., Leighliter, B., Morton, J. T. & Lenhart, J. L. *Smart Mater. Struct.* **25**, 025004 (2016).

2. In a simple melt, polymer chains have physical crosslinks in the form of entanglements. In these systems, few entanglements have been replaced by permanent crosslinks. So, I don't understand the idea of polymer being confined. I don't think polymer can be considered to be confined, unless they deviate from their equilibrium conformation. Is it so?

Response: We appreciate the reviewer's helpful suggestion. The confine in our paper refers to the limitation of the relaxations of polymer chains in the crosslinked network. This reason is that the molecular weight of cross-linking strands (M_c) is less than that of entanglement in the network (M_e). The experimental data and theoretical analysis (Fig. 3a) also show that the whole chain time of PBA fluids in a fixed network is longer than that of PBA melts.

3. Authors claim that the damping originates from the frictional losses of free polymer chains during their reptation. Based on this claim, damping should always be more for the higher Mw free polymers. Authors also claim that damping occurs for the region where tan delta is maximum, which is not same as having higher frictional losses.

Overall, the mechanism and the explanation of damping remains poorly discussed.

Response: We thank the reviewer for this comment. We do not think damping should always be more for the higher M_w free polymers. As the M_w of polymer fluids increases, their reptation time becomes longer, resulting in more frictional losses. However, with the same polymer mass, the longer the polymer chains, the less the number of chains. The frictional losses do not increase with increasing M_w of the polymer fluids. This situation is similar to the entanglement modulus of the polymers ($G_e \cong \rho RT / M_e$), which does not vary with the molecular weight.

The damping mechanism of the PFGs: in the PFGs, the polymer fluids in the viscous state offer high internal friction by the chain reptation, leading to a dramatic increase of their energy dissipation.

4. I don't see any significant advantage of the material being a single component. Miscibility could be a concern, but a combination of polymeric entities which have negative Flory-Huggins interaction parameter can be used to ensure miscibility.

Response: We thank the reviewers for this comment. The significant advantages of the polymer materials with a single component are as follows:

a. Although we can choose materials with low value of Flory-Huggins interaction parameter (χ) to enhance the miscibility, it is actually not easy to guarantee the true miscibility for a multiple-phase system over an extremely broad temperature range, and the system becomes immiscible when the molecular weight is large.

b. The polymer blends with negative χ usually have supramolecular interactions, such as hydrogen bond interaction and π - π interaction. These interactions result usually in the LCST behaviors for polymer blends. These phase transition behaviors need to be avoided for the application of polymer materials as that decreases the application temperature range of materials.

Therefore, to ensure that phase separation does not occur over a broad range of temperature, we chose single-component materials as the research object.

5. Introduction focuses on damping via frictional losses. How is that related to

stretchability discussed in results section?

Response: We thank the reviewers for this comment. As the PFGs are a new type of the damping materials, it is necessary to investigate the mechanical properties systematically. The stretchability of materials is one of the important mechanical properties. The PFGs exhibit the desirable stretchability due to the existence of polymer fluids, which is related to the topic of this article.

Besides these major issues there are several minor issues. Here, I will just mention a few for authors' benefit:

a. What are PFG 5 and 6? Why aren't they not mentioned in table 1?

Response: We appreciate the reviewer's helpful suggestion. In our revised manuscript, we have added Table 1 including all sample parameters.

b. There are several grammatical issues with the writing and construction of statements is poor.

Response: We thank the reviewer for this helpful suggestion. In our revised manuscript, we have modified the grammar and presentation problems.

c. Why have the authors chosen a flexible polymer? Why PBA?

Response: We thank the reviewer for this helpful suggestion. Flexible polymers possess low T_g , and they are in the flow state at room temperature. PBA is a kind of prototypical and common flexible polymers. PBA is chosen as the research object, which has stronger universality and persuasiveness.

d. What is the state of the art stretchability of other polymer-based materials?

Response: We thank the reviewer for this suggestion. The state of the art stretchability of other polymer-based materials can be divided into two types: physical dynamic cross-linking and chemical covalent cross-linking. The polymer materials with the physical dynamic cross-linking exhibit a maximum elongation of 10,000% (Li, C. *et*

al. Nature Chemistry 2016, 8, 618-624). It arises from the continuous chain slippage in the polymer chain during stretching. Thus, this deformation is irreversible. The polymer materials with chemical covalent cross-linking exhibit a maximum elongation of ~500%, owing to the inherent entanglement the polymer chains (Vatankhahvarnosfaderani, M. *et al. Nature* **549**, 497–501 (2017)).

e. What do the authors mean by van der Waals interaction being ‘favorable’?

Response: We thank the reviewer for this suggestion. This means that there is a strong intermolecular interaction in the PFGs, which endows them excellent tensile properties and robust fatigue resistance.

f. Why is the x-axis of figure 2j $M_n^{1/2}$ instead of just M_n ? Is the data on Y axis $\tan(\delta)$ or peak $\tan(\delta)$?

Response: We thank the reviewer for pointing this out. We have changed $M_n^{1/2}$ to M_n in the x-axis of fig. 2j in the revised manuscript. The data on Y axis is the value of $\tan(\delta)$ at a frequency of 10 rad/s and $T = 25$ °C.

Reviewers' Comments:

Reviewer #1:

Remarks to the Author:

The authors have, by and large, satisfactorily addressed the points that I raised in my first review. I do however want to make a general point concerning their response to my comments, which should be addressed before the ms. is accepted.

The point is that the authors address several of the points BY EXPLANATIONS IN THEIR RESPONSE LETTER RATHER THAN BY REVISING THE MS., which is what they should have done. In addition, even for those points where they do apparently revise the ms., they do not provide the revised text in their response letter, which makes it difficult to see what they have done.

Since the points I raised were not for my own satisfaction but to clarify to the readers, it is essential that the authors actually revise the ms., if only briefly, either the main text or the SI, in response to each comment, and not just provide an explanation for the reviewers in the response letter.

Thus for example to address my point 2, they must indicate what revision they make in the text or the SI, and provide that revised text as part of the response to Comment 2. Also, where do figures R1, R2 etc. appear? If they are just for the response letter, that is not helpful to the reader: they must be referred to in the main text and be added to the SI. This is the usual way to respond to reviewers.

Likewise for my comment 4: a brief revision to the text must be made, summing up the explanation they provide in the response letter, and this revised text should be provided as part of their response to comment 4, etc.

And so on for the other points. Where they have already revised the text, as in their response to my point 7, for example, they should provide this revision in the response letter, rather than forcing the reviewer to chase up the revisions in the text.

However, I should emphasize, as I wrote in my first review, that their study is novel and interesting (and the raw egg dropping video is neat!), and once the authors have revised the text and SI of their ms. in line with the satisfactory explanations that they have already provided in their response letter, I would definitely recommend that it be published in Nature Communications.

Reviewer #2:

Remarks to the Author:

The authors have largely addressed my comments. However, the authors should include in the manuscript the justification on why PFGs are better than the 102K PBA, even when the response of the 102k PBA fluid itself is close to the response of the mixed PFG. Following this addition, and a further thorough editing of the manuscript to eliminate factual, typographical, and grammatical errors, the publication of this manuscript in Nature Communications is recommended.

-Samanvaya Srivastava

Reviewer #3:

Remarks to the Author:

I thank the authors for revising the manuscript and for their efforts to address my concerns. After going through the rebuttal letter and revised manuscripts, following points remain my major concern. Unless these concerns are correctly addressed, I am not in favor of publication of the manuscript.

a. The Introduction still remains weak. The exact requirement of an ideal damping material is

missing. With the help of some examples authors should describe the exact need and the corresponding desired frequency range. Just stating "high performance damping materials over a broad frequency range" is vague.

b. I still maintain my argument, that for higher Molecular weight polymer samples, frictional losses will be higher. Plateau modulus represents energy density stored in the material, whereas frictional losses are not to be evaluated per volume.

c. I would again argue that a cross-linked polymer should not be considered/called as confined system.

d. What I can understand is that these PFGs are a slight variation to L-gels. So that claim these PFGs are novel should be nuanced.

Reviewers' comments:

Reviewer #1 (Remarks to the Author):

The authors have, by and large, satisfactorily addressed the points that I raised in my first review. I do however want to make a general point concerning their response to my comments, which should be addressed before the ms. is accepted. The point is that the authors address several of the points by explanations in their response letter rather than by revising the ms., which is what they should have done. In addition, even for those points where they do apparently revise the ms., they do not provide the revised text in their response letter, which makes it difficult to see what they have done. Since the points I raised were not for my own satisfaction but to clarify to the readers, it is essential that the authors actually revise the ms., if only briefly, either the main text or the SI, in response to each comment, and not just provide an explanation for the reviewers in the response letter.

1. Thus for example to address my point 2, they must indicate what revision they make in the text or the SI, and provide that revised text as part of the response to Comment 2. Also, where do figures R1, R2 etc. appear? If they are just for the response letter, that is not helpful to the reader: they must be referred to in the main text and be added to the SI. This is the usual way to respond to reviewers.

2. Likewise for my comment 4: a brief revision to the text must be made, summing up the explanation they provide in the response letter, and this revised text should be provided as part of their response to comment 4, etc.

3. And so on for the other points. Where they have already revised the text, as in their response to my point 7, for example, they should provide this revision in the response letter, rather than forcing the reviewer to chase up the revisions in the text.

However, I should emphasize, as I wrote in my first review, that their study is novel and interesting (and the raw egg dropping video is neat!), and once the authors have revised the text and SI of their ms. in line with the satisfactory explanations that they have already provided in their response letter, I would definitely recommend that it be

published in Nature Communications.

Reviewer #2 (Remarks to the Author):

The authors have largely addressed my comments. However, the authors should include in the manuscript the justification on why PFGs are better than the 102K PBA, even when the response of the 102k PBA fluid itself is close to the response of the mixed PFG. Following this addition, and a further thorough editing of the manuscript to eliminate factual, typographical, and grammatical errors, the publication of this manuscript in Nature Communications is recommended.

-Samanvaya Srivastava

Reviewer #3 (Remarks to the Author):

I thank the authors for revising the manuscript and for their efforts to address my concerns. After going through the rebuttal letter and revised manuscripts, following points remain my major concern. Unless these concerns are correctly addressed, I am not in favor of publication of the manuscript.

- a. The Introduction still remains weak. The exact requirement of an ideal damping material is missing. With the help of some examples authors should describe the exact need and the corresponding desired frequency range. Just stating "high performance damping materials over a broad frequency range" is vague.
- b. I still maintain my argument, that for higher Molecular weight polymer samples, frictional losses will be higher. Plateau modulus represents energy density stored in the material, whereas frictional losses are not to be evaluated per volume.
- c. I would again argue that a cross-linked polymer should not be considered/called as confined system.
- d. What I can understand is that these PFGs are a slight variation to L-gels. So that claim these PFGs are novel should be nuanced.

Reviewer #1

The authors have, by and large, satisfactorily addressed the points that I raised in my first review. I do however want to make a general point concerning their response to my comments, which should be addressed before the ms. is accepted. The point is that the authors address several of the points by explanations in their response letter rather than by revising the ms., which is what they should have done. In addition, even for those points where they do apparently revise the ms., they do not provide the revised text in their response letter, which makes it difficult to see what they have done. Since the points I raised were not for my own satisfaction but to clarify to the readers, it is essential that the authors actually revise the ms., if only briefly, either the main text or the SI, in response to each comment, and not just provide an explanation for the reviewers in the response letter.

Response: We thank the reviewer for the highly recognition and constructive suggestions of our work. We have revised the manuscript. Revisions in the main text, Supplementary Information (SI) file and response letter are highlighted.

1. Thus for example to address my point 2, they must indicate what revision they make in the text or the SI, and provide that revised text as part of the response to Comment 2. Also, where do figures R1, R2 etc. appear? If they are just for the response letter, that is not helpful to the reader: they must be referred to in the main text and be added to the SI. This is the usual way to respond to reviewers.

Response: We appreciate the reviewer's comments. In response to this comment, we have added the content of comparison with the damping gel (Taica Company) in the revised manuscript.

Damping property: we have added the temperature range of $\tan\delta > 0.5$ of the damping gel in the revised main text (Fig. 3c) and Supplementary Table S4. We also added the source of the damping gel data in the revised Supplementary Information (P. 26, L. 150).

“[8]. https://taica.co.jp/gel/en/product/shock_absorption/theta_sheet.html”

Figure 3 | (c) The temperature range of $\tan\delta > 0.5$ of the PFGs compared to the reported damping materials.

Table S4. Temperature range of $\tan\delta > 0.5$ of the reported damping materials.

Damping materials	$\tan\delta > 0.5$ (°C range)	$\tan\delta > 0.5$ (°C width)	References
Chlorinated butyl rubber	-57 to 8	65	(1)
Styrene-butadiene rubber	-43 to -20	23	(2)
Gradient PEG	-30 to 18	48	(3)
NBR/AO-80 blend	3 to 35	32	(4)
Thiol-ene polymer	10 to 42	32	(5)
P(BA-co-MMA)	26 to 71	45	(6)
PU/EP IPN	38 to 100	62	(7)
Damping gel	-90 to -10	80	(8)

The temperature range is calculated from the frequency-dependent $\tan\delta$ curves in the references.

Shock absorption: we have added the drop weight test data of the damping gel in the Supplementary Fig. 17 and main text (P. 12, L. 253–254).

“Compared to other commercial damping materials like the silicone rubber and damping gel, the PFG_(3%, 60%-35k) also exhibits a lower rebound.” (P. 12, L. 253–254)

Figure S17. Drop weight impact tests on the PFG_(3%, 60%-35k) and the damping gel.

Vibration absorption: the vibration absorption capacity of the damping gel was measured by shaking table demonstrative experiments. The experimental data has been added in the revised Supplementary Information (Supplementary Fig. 19) and main text (P. 12, L. 258–260).

“Moreover, the vibration absorption capacity of the PFG_(3%, 60%-35k) was measured by shaking table demonstrative experiments (Fig. 4d and Supplementary Fig. 19).” (P. 12, L. 258–260)

Figure S19. Shaking table demonstrative experiments of the PFG_(3%, 60%-35k) and the damping gel.

Egg dropping experiments: we demonstrated the egg was dropped from 2 m height onto the 5 mm-thick damping gel. We have added this result in the revised main text (P. 12, L. 257–258) and Supplementary Movie 5.

“while dropped onto the natural rubber pad and damping gel with serious rupture (Supplementary Movie 4 and Movie 5).” (P. 12, L. 257–258)

Acoustic absorption: The damping gel with 5mm height was placed under the petri dishes containing sands and put on an ultrasonic generator. This result has been added in the main text (P. 12, L. 262–263 and 264–265) and Supplementary Movie 7.

“Meanwhile, we tested the acoustic sound absorption of the PFG_(3%, 60%-35k) using an ultrasonic generator (Fig. 4e, Supplementary Movie 6 and Movie 7),” (P. 12, L. 262–263)

“When the ultrasound wave with a frequency of 1 MHz is applied, the sands atop the natural rubber (NR) and damping gel spacer beat violently.” (P. 12, L. 265–266)

2. Likewise for my comment 4: a brief revision to the text must be made, summing up the explanation they provide in the response letter, and this revised text should be provided as part of their response to comment 4, etc.

Response: We appreciate the reviewer’s helpful question. In response to the comment 4, we have added the following explanation in the main text (P. 6, L. 124–131).

“In addition, there are other reasons for the ultrahigh stretchability of the PFG. At 50-fold extension, some network links break to give this huge extension. This is in line with the 250% residual strain, which points to irreversible chain scission following the 3,000% extension (Fig. 2c) of PFG_(0.1%, 40%-5k). The other point is that the regular network model is assumed in the theoretical estimation of the maximum elongation, while the PFG network is irregular. Polymer networks are formed via the free radical polymerization process. Consequently, they possess topological defects, which can improve the tensile rate to some extent.” (P. 6, L. 124–131)

3. And so on for the other points. Where they have already revised the text, as in their

response to my point 7, for example, they should provide this revision in the response letter, rather than forcing the reviewer to chase up the revisions in the text.

Response: We thank the reviewer for this suggestion. In response to the comment 7, the characterization of linear viscoelastic region for the PFG has been added to the revised Supplementary Material (Supplementary Fig. 5). We have also added the following explanation and experimental methods in the main text (P. 7, L. 144–146 and P. 15, L. 314–316).

“The PFG is in the linear viscoelastic region within 10% of the strain (Supplementary Fig. 5), which indicates the rheological data measured at 0.5% strain are reversible.” (P. 7, L. 144–146)

“The samples were placed under a 15 mm-diameter parallel plate. In the strain sweep tests, the shear strain (γ) was from 0.01% to 10% at the frequency of 10 rad/s and temperature of 25 °C.” (P. 15, L. 314–316)

Figure S5. The shear strain (γ) of storage moduli (G') and loss moduli (G'') for PFG_(3%, 60%-35k) were obtained at the frequency of 10 rad/s and temperature of 25 °C.

However, I should emphasize, as I wrote in my first review, that their study is novel and interesting (and the raw egg dropping video is neat!), and once the authors have revised the text and SI of their ms. in line with the satisfactory explanations that they have already provided in their response letter, I would definitely recommend that it be published in Nature Communications.

Response: We thank the reviewer for the highly recognition and constructive suggestions of our work.

Reviewer #2 (Remarks to the Author):

The authors have largely addressed my comments. However, the authors should include in the manuscript the justification on why PFGs are better than the 102K PBA, even when the response of the 102k PBA fluid itself is close to the response of the mixed PFG. Following this addition, and a further thorough editing of the manuscript to eliminate factual, typographical, and grammatical errors, the publication of this manuscript in Nature Communications is recommended.

-Samanvaya Srivastava

Response: We thank the reviewer for the positive evaluation and constructive suggestions of our work. In response to this comment, we have added the following justification of using a mixture as the liquid phase in the main text (P. 11, L. 225–232).

“Compared to the PFG with a single polymer fluid, the damping performance of PFG-b1 exhibits higher stability and controllability, though the high damping ($\tan\delta > 0.5$) frequency range of the PFG with a single polymer fluid (such as $M_n = 102k$) is close to that of the PFG-b1 under certain circumstances (Fig. 3b). Owing to the gradual relaxation of various PBA fluids in the PFG-b1, the $\tan\delta$ curve of PFG-b1 displays a more stable trend rather than a pronounced peak (Fig. 3b), and the PFG-b1 can achieve more possibilities for the damping performance by regulating the PBA fluid ratio.” (P. 11, L. 225–232)

Reviewer #3 (Remarks to the Author):

I thank the authors for revising the manuscript and for their efforts to address my concerns. After going through the rebuttal letter and revised manuscripts, following points remain my major concern. Unless these concerns are correctly addressed, I am not in favor of publication of the manuscript.

Response: We thank the reviewer for the positive evaluation and helpful suggestions.

a. The Introduction still remains weak. The exact requirement of an ideal damping material is missing. With the help of some examples authors should describe the exact need and the corresponding desired frequency range. Just stating "high performance damping materials over a broad frequency range" is vague.

Response: We thank the reviewer for this comment. In response to these comments, we have added the following examples and their associated frequency ranges in the revised main text (P. 2, L. 32–36).

“Vibrations and noises of various frequency bands are ubiquitous in various engineering fields. For instance, the vehicles¹, aircrafts² and noises³ are the common vibration sources with diverse associated frequency ranges of $10^0 \sim 10^4$, $10 \sim 10^2$, and $10^{-1} \sim 10^6$ Hz, respectively. These vibrations can cause malfunctioning, resonance, or fatigue failure of critical structures and human injury^{4,5}.” (P. 2, L. 32–36)

We have also cited the related papers in the revised main text.

“[1] Qatu, M., Abdelhamid, K., Pang, J. & Sheng, G. *Int. J. Vehicle Noise and Vibration* **5**, 1–35 (2009). [2] Fidell, S., Pearsons, K., Silvati, L. & Sneddon, M. *J. Acoust. Soc. Am.* **111**, 1743–1750 (2002). [3] Berglund, B., Hassmen, P. & Job, R.F.S. *J. Acoust. Soc. Am.* **99**, 2985–3002 (1996).” (P. 17, L. 358–364)

b. I still maintain my argument, that for higher Molecular weight polymer samples, frictional losses will be higher. Plateau modulus represents energy density stored in the material, whereas frictional losses are not to be evaluated per volume.

Response: We appreciate and agree with the reviewer’s comments. When the polymer

chains move at the same time or temperature scale, the higher the molecular weight of the polymer chains is, the higher the frictional losses are. However, with increasing the molecular weight of polymer fluids, the reptation time of polymer fluids becomes longer, indicating that their reptation occurs at lower frequencies or higher temperatures. In this case, local chain interactions become lower, leading to a decrease of the frictional losses of their reptation (Shaw, M. T. & Macknight, W. J. *Introduction to Polymer Viscoelasticity* (Wiley, New York, 1983)). This could cause the decrease of the damping performance with the increase of molecular weight of polymer fluids.

c. I would again argue that a cross-linked polymer should not be considered/called as confined system.

Response: We appreciate the reviewer's comment. The confinement in our paper refers to the limitation of the movement of polymer chains in the crosslinked network. In a polymer melt with molecular weight higher than the entanglement molecular weight (M_e), the mechanism is the well-known reptation model: the long-chain polymer undergoes diffusion by an one-dimensional random walk along the tube, created by the entanglement and which diameter is about $\sqrt{M_e}b$. In our experiments, since the molecular weight of cross-linking strands (M_c) is less than M_e , thus the mesh size is less than the tube diameter. In this case, the movement of free polymer chains is restricted by smaller mesh size. In this sense, we call a cross-linked polymer network as a confined system. Similar presentation had been used (Jia, D. & Muthukumar M. *Nat. Commun.* **9**, 2248 (2018)), where the authors studied the dynamics of a long polyelectrolyte chain in a hydrogel.

d. What I can understand is that these PFGs are a slight variation to L-gels. So that claim these PFGs are novel should be nuanced.

Response: We appreciate the reviewer's comments. We would like to clarify that the PFGs are entirely different from the L-gels. The comparisons of the PFGs and the

L-gels are as follows:

Table R1. The comparison of the PFGs and the L-gels.

	Component	Disperse medium	Damping mechanism	Damping properties ($T = 25\text{ }^{\circ}\text{C}$)	Damping controllability
L-gel	Network + Small molecules	Small molecules	Chain segment relaxation	Tan < 0.1 ($10^{-2} \sim 10^0$ Hz)	No
PFG	Network + Polymer fluids	Polymer fluids	Whole chain relaxation	Tan < 0.5 ($10^{-2} \sim 10^8$ Hz)	Yes

Components: The L-gels consist of polymer networks swelled by small molecules, which belong to the traditional organogels. The PFGs consist of polymer networks infused by polymer fluids. Obviously, the components of PFGs are fundamentally different from that of L-gels. For this reason, the damping mechanism and properties of the PFGs and L-gels are also absolutely different.

Damping mechanism: in the L-gels, the physical cross-linked chains (thermally reversible) significantly outnumber the covalent cross-links (permanent). Upon application of heat, the physical cross-links dissociate resulting in an increased number of dangling chain ends, where the dangling chain ends promote energy dissipation to provide enhanced mechanical damping. This mechanism of utilizing the segment motion of dangling chains to realize energy dissipation is also summarized in the introduction section (L. 45).

Our works (PFGs) utilize the internal friction arising from the whole chain relaxation of the confined fluid in the network to realize energy dissipation.

Damping properties: At $25\text{ }^{\circ}\text{C}$, the loss factor ($\tan\delta = G'' / G' < 0.1$) of L-gels damping coefficient of the polymer is very low, which is similar to that of the liquid gels. Only when the temperature is above $60\text{ }^{\circ}\text{C}$, the L-gels dissociate show excellent damping performance, due to the dissociation of their physical cross-links. These gels

focus on mechanical switching according to the local environment rather than the tunable damping performance.

The PFGs' energy-dissipation property can be precisely tailored at desired frequencies through adjusting the relaxation time of polymer fluids by varying chain length. Moreover, the PFGs exhibit high energy-dissipation performance (loss factor > 0.5) with a broad frequency ($10^{-2} \sim 10^8$ Hz)/temperature (-50 ~ 120 °C) by infusing several polymer fluids with significantly different chain length.

Reviewers' Comments:

Reviewer #1:

Remarks to the Author:

The authors have addressed the points in my second review and I can recommend that the ms. be published in Nature Communications.

Reviewer #2:

Remarks to the Author:

I am satisfied with the revisions that the authors have made. I recommend the publication of this manuscript in Nature Communications.

Reviewer #3:

None

The last referee #3 comments:

a. The Introduction still remains weak. The exact requirement of an ideal damping material is missing. With the help of some examples authors should describe the exact need and the corresponding desired frequency range. Just stating "high performance damping materials over a broad frequency range" is vague.

Response: we have added the following examples and their associated frequency ranges in the revised main text.

“Vibrations and noises of various frequency bands are ubiquitous in various engineering fields. For instance, the vehicles¹, aircrafts² and noises³ are the common vibration sources with diverse associated frequency ranges of $10^0 \sim 10^4$, $10 \sim 10^2$, and $10^{-1} \sim 10^6$ Hz, respectively. These vibrations can cause malfunctioning, resonance, or fatigue failure of critical structures and human injury^{4,5}.” (P. 2, L. 33–37)

We have also cited the related papers in the revised main text.

“[1] Qatu, M., Abdelhamid, K., Pang, J. & Sheng, G. *Int. J. Vehicle Noise and Vibration* **5**, 1–35 (2009). [2] Fidell, S., Pearsons, K., Silvati, L. & Sneddon, M. *J. Acoust. Soc. Am.* **111**, 1743–1750 (2002). [3] Berglund, B., Hassmen, P. & Job, R.F.S. *J. Acoust. Soc. Am.* **99**, 2985–3002 (1996).” (P. 17, L. 372–378)

b. I still maintain my argument, that for higher Molecular weight polymer samples, frictional losses will be higher. Plateau modulus represents energy density stored in the material, whereas frictional losses are not to be evaluated per volume.

Response: we have added the following justification in the main text.

“With increasing M_n (PBA fluid), the peak value of $\tan\delta$ decreases. It is due to the fact that the reptation time of polymer fluids becomes longer, indicating that their reptation occurs at lower frequencies or higher temperatures. In this case, local chain interactions become lower, leading to a decrease of the frictional losses of their reptation.” (P. 9, L. 183–186)

c. I would again argue that a cross-linked polymer should not be considered/called as confined system.

Response: we have added the following content in the main text.

“In the PFGs, since the molecular weight of cross-linking strands ($M_x = 4.2$ kDa) is less than the entanglement molecular weight ($M_e \cong 20$ kDa)³⁴, the mesh size is less than the tube diameter. In this case, the movement of the polymer fluid chains is restricted by smaller mesh size.” (P. 10, L. 215–218)

d. What I can understand is that these PFGs are a slight variation to L-gels. So that claim these PFGs are novel should be nuanced.

Response: we have added the following content in the main text.

“Traditionally, polymer networks are infused by small molecules to form liquid gels, such as hydrogels or organogels. The infusion of a fluid phase into the polymer networks will significantly enhance the chain relaxation and reduce the internal frictions, leading to an inferior damping property²⁹⁻³¹. Even although these gels can improve the damping performance by using phase transition behavior, this behavior only occurs at certain temperature conditions and makes their mechanically unstable²⁵.” (P. 4, L. 74–79)

Response to referees' comments:

Reviewer #1 (Remarks to the Author):

The authors have addressed the points in my second review and I can recommend that the ms. be published in *Nature Communications*.

Response: We thank the referee for the encouraging comments and helpful suggestions. We are glad to have the referee's recommendation for publication in *Nature Communications*.

Reviewer #2 (Remarks to the Author):

I am satisfied with the revisions that the authors have made. I recommend the publication of this manuscript in *Nature Communications*.

Response: We thank the referee for the high recognition of our work. We are glad to have the referee's recommendation for publication in *Nature Communications*.